



# Evaluation of the impact of wind farm control techniques on fatigue and ultimate loads

Alessandro Croce[1], Stefano Cacciola[1], Luca Sartori[1], and Paride De Fidelibus[1]

[1]Department of Aerospace Science and Technology, Politecnico di Milano, Milano, Italy

**Correspondence:** Alessandro Croce (alessandro.croce@polimi.it)

**Abstract.** Wind farm control is one of the solutions recently proposed to increase the overall energy production of a wind power plant.

A generic wind farm control is typically synthesized so as to optimize the energy production of the entire wind farm by reducing the detrimental effects due to wake-turbine interactions. As a matter of fact, the performance of a farm control is

typically measured by looking mainly at the increase of produced power, possibly weighted with the wind Weibull and rose at a specific place, and, sometimes, by looking also at the fatigue loads. However, an aspect which is rather overlooked is the evaluation of the impact that a farm control law has on the maximum loads and on the dynamic responses under extreme conditions of the individual wind turbine.

In this work, two promising wind farm controls, based respectively on Wake Redirection (WR) and Dynamic Induction

Control (DIC) strategy, are evaluated at a single wind turbine level. To do so, a two-pronged analysis is performed. Firstly, the control techniques are evaluated in terms of the related impact on some specific key performance indicators (e.g. fatigue and ultimate loads, actuator duty cycle and annual energy production). Secondarily, an optimal blade redesign process, which takes into account the presence of the wind farm control, is performed with the goal of quantifying the possible modification in the structure of the blade and hence of quantifying the impact of the control on the Cost of Energy model.

## 1 Introduction and motivation

So far, the majority of the works devoted to wind farm control are aimed at evaluating the effectiveness of such techniques as means of power harvesting maximization. Among all, one can mention the methodologies based on wake steering (see Fleming et al. (2019); Gebraad et al. (2017, 2016)), steady axial induction (see Annoni et al. (2016)) and dynamic induction control, also called active wake mixing (see Munters and Meyers (2018, 2017)). As long as the energy production is viewed as the most

significant merit figure, these analyses considered only the overall farm annual energy production (AEP) and, only in some cases, the fatigue loads on the single wind turbine (Bossanyi, 2018; Knudsen et al., 2015).

Operating according to a farm control may force a turbine to work in sub-optimal regimes, e.g. with large misalignment or with dynamically varying thrust. In such conditions, it is possible to experience significant increases not only in the turbine fatigue, (Cardaun et al., 2019; Ennis et al., 2018; White et al., 2018), but also in the ultimate loads or maximum blade deflections,

which typically come from extreme event, as gusts or faults.



Fatigue, ultimate loads and maximum tip displacements participate together in the definitions of the constraints to which a machine is subject during the design phase. Hence, the possible increase in machine loading induced by wind farm control determines whether a turbine structure is to be re-designed with an eventual increase of its mass and cost or not.

As expected, dealing with a farm control, its advantages (increased power production at farm level) and disadvantages (possible increased loading at turbine level) combine together to give the cost a farm has to produce an amount of energy. In order to show this concept, one may consider the simple definition of the cost of the energy (CoE) of a single turbine, reported in Fingersh et al. (2006),

$$\text{CoE} = \frac{\text{FCR} \cdot \text{ICC}}{\text{AEP}} + \text{AOE},$$

where FCR is the fixed charge rate, ICC the initial capital cost, consisting basically in the turbine cost, and AOE the annual operating expenses, which may include land or sea lease and operation and maintenance costs. Clearly, the AEP of a single turbine may be reduced or increased by the farm control according to the fact that a machine operates mainly upstream or downstream. The sum of the AEPs of all turbines belonging to the farm results increased for a farm control neatly designed. Besides that, the related effect on the turbine loading and, eventually, on ICC, is still to be carefully determined in order to find the final effect on CoE.

To do so, given a specific wind farm control techniques, the component loading should be evaluated on a turbine level, so as to clarify if a generic turbine within a "controlled farm" would need a dedicated design and, consequently, evaluate the effect of the farm control in terms of ICC.

The scope of this paper is twofold. First, to quantify the impact of two wind farm control algorithms, i.e. wake steering and dynamic induction control, at turbine level through the evaluation of fatigue and ultimate loads and maximum blade tip deflection. Second, to quantify the possible increase in the blade mass, taken as a measure of the increase in the cost of the rotor, when an optimal blade design is performed including the presence of the aforementioned farm control algorithms. All analyses are performed on a model of a 10 MW wind turbine, which can be considered as a generic reference for future machines proposed for the exploitation of on- and off-shore resources.

The present paper is organized according to the following plan. Section 2 and 3 deal with the explanation of the methodology adopted to evaluate the impact of wind farm control on single wind turbine level. These sections include the description of the considered machine and its controller, of the multibody software used for the aeroservoelastic model and analyses and of the optimal rotor design tool. In Sec. 4, a sensitivity analysis on the effects of two wind farm control techniques (i.e. WR and DIC) is considered. Specifically, fatigue and ultimate loads, along with maximum blade tip deflection, are evaluated considering different setting of the control techniques (e.g. different yaw misalignment angles for wake redirection technique) in order to find the most impacting conditions for the turbine. Such sensitivity analysis is viewed as a preliminary step for the optimal blade design process which is described in Sec. 5. Finally, Sec. 6 finalizes the manuscript by listing the main findings and possible outlooks of the work.





## 2 Methodology

Having a global vision on the effects that a farm control has on turbine and eventually quantifying its impact on the design of
blades is not an easy task. In fact, when wind farm control is of concern, the problem of analyzing wind turbine performance
becomes highly site specific as the inputs of the farm control to the single turbines will depend on many factors, such as the
farm geometry, the wind distribution and rose, the turbulence intensities. In such a scenario, deriving conclusions of general
validity without focusing too much on a specific case is a difficult task.

Moreover, in a same wind farm, a turbine may act according to a farm control input (e.g. operates at a specific yaw misalign-
ment angle to redirect its wake), or feel the effects of the control action performed by another machine, depending on whether
it is up- or down-stream with respect to the wind direction. As a consequence, one should have to model all possible cases to
have a global overview. This clearly poses some difficulties as the study would results to be again strongly dependent on the
wind farm geometry and would lose, at least in part, its generality.

Last but not least, modern wind turbine are designed according to international Standards, which prescribe the computation
of fatigue and ultimate loads in a certain number of conditions, e.g. for specific wind speeds and turbulence intensity levels.
Hence, the impact of a farm control should be evaluated through those Standards prescriptions. Unfortunately, regulations, in
their current status, do not consider the fact that a turbine may operate out of the design conditions according to a farm control.

This discussion highlights the fact that there are three critical issues in this study which should be neatly addressed: site-
dependency of the problem, if and how to consider the effects of wind farm control on downstream machines and inclusion of
the farm control within the Standards.

To overcome the first point, in this work, all analyses have been conducted as sensitivity studies, in which the effects of wind
farm control are considered as functions of some important parameters. Hence, the study of the wake redirection was carried
out for different values of yaw misalignment, whereas that of the DIC for different frequency and amplitude of the pitch
oscillation. From this point of view, such a sensitivity analyses can be even used as input for the synthesis and fine tuning of a
farm control by reducing its authority in those conditions which could be critical from the loading sides. Some example can be
listed. One may be interested in limiting yaw misalignment angle in a range of the machine envelope where a possible extreme
events, a gust or a fault may cause an exceptional increase of an design load. Otherwise, the amplitude and the frequency of
the pitch motion for dynamic induction control can be chosen also by looking at the effect of fatigue and actuator duty cycle.
This possibility, although interesting, is not considered in this paper, as it is out of its scope, and will be further investigated as
a follow-up of this work.

Dealing with the effects of the farm control on downstream machines, it is important to stress the fact that real turbines
typically operate in a farm and are certainly involved in wake interaction phenomena. Notwithstanding that, they eventually
arrive to the end of life with a prospective life residual to be exploited (Ziegler et al., 2018). This said, one could conclude that
modern design procedures and regulations are adequate to guarantee safe wind turbine operations even in presence of normal
wake impingement events. For this reason, the analyses conducted in this work refer only to the upstream turbine, i.e. the one



which performs an action according to the control of the farm; an aspect which is actually not considered in current design procedures.

The discussion about the Standards deserves a special attention. To this end, consider a simplified wind farm made by three INNWIND.EU 10 MW turbines. Figure 1 on left represents a simulated flow within such a wind farm on a plane parallel to the
ground at hub height. The flow has been simulated with the open-source version of Floris (FLOw Redirection and Induction in Steady State) written in Python (see NREL (2019)). The wind, 8 $m/s$ with turbulence intensity of 2%, is coming 35 deg North-West and, as visible from the upper plot, generates strong wake impingement between the first and the second turbine, and between the second and the third. In this situation it is possible to increase the power harvested by the farm by yawing the turbines so as to redirect their wakes as demonstrated by the bottom plot. The yaw misalignment angles of the first two turbines
have been computed in order to optimize, within the limits of this stationary model, the total farm power production obtaining the optimal setpoints of yaw angles in this case, i.e about -20 and 24 deg of yaw respectively. The very same analysis and optimization have been carried out for different wind speeds and different turbulence intensity levels. Figure 1, on the right, shows the contour plot of the optimal yaw angle of the first turbine. Clearly, for low speed and low turbulence intensity the turbine has to yaw of significant yaw angles (higher than 20 deg) to optimize the energy production. On the other hand, as the
wind speed increases the optimal yaw decreases up to zero in the full power region, where actually the farm control results not necessary. Additionally, since higher turbulence levels are responsible for an increased in-wake flow mixing and in turn faster wake dissipation, the yaw angle decreases also as TI increases. The dark blue line refers to a yaw angle of 2 deg and, hence, can be viewed also as the boundary between the two regions where the farm control is active or not. Superimposed to this contour, the plot displays also the conditions prescribed by the Standards IEC-6400 for the computation of the Design Load Cases
(DLC) for turbine class $I_A$(IEC 61400-1 Ed.3., 2004), as black lines and triangles. Just at a first sight, one may recognize that the majority of the conditions considered in the Standards seems out of the region of activation of the control, especially those at Extreme Turbulence Model (ETM) and Nominal Turbulence Model (NTM), the latter used for fatigue calculation. Extreme Wind Shear (EWS), Extreme Change of Direction (ECD) and Extreme Operating Gust (EOG), on the other side, refer to steady winds and are associated to an ideal null TI. In this situation, the analysis appears extremely complicated, especially because
of the dependency of the optimal yaw angle from the TI, which, with the current state of the regulation, would practically exclude the farm control from NTM and hence from fatigue and AEP calculation. Moreover, a further question arises over the actual implementation of the control, on whether it can rely on a good measure of TI and is synthesized so as to account for it. This stresses two facts. First, including a TI-dependency of the control in this analysis is problematic and second, an update of the Standards to consider wind farm control is in any case necessary. The latter issue, although important and interesting
is out of the scope of the paper. Dealing with the TI-dependency, in order to provide an analysis of general validity and to simplify the treatment, the work of this paper considered only a wind speed-dependency of the control and imposed as limit for its activation 15 m/s no matter of TI. Clearly, this assumption appears strong but it allows one to focus directly on the worst scenario and hence to show the maximum possible impact of the wind farm control on turbine loading.

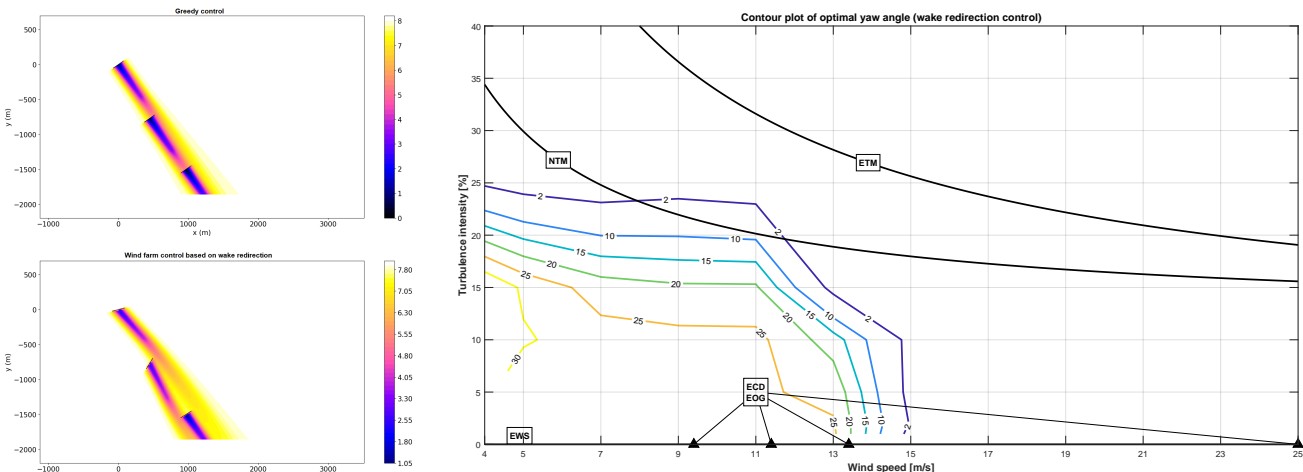

**Figure 1.** Investigation of wake redirection technique as function of wind speed and TI. Left plot, Floris simulation of three-turbine farm at 8 m/s and TI 2%. Right plot: contour of optimal yaw angles of the first turbine superimposed to the conditions prescribed by Standards for design load cases (DLC) calculation.

## 3 Tools of analysis

### 3.1 Wind turbine model and simulation environment

The wind turbine considered in this work is the reference INNWIND.EU model (DTU, 2012), which is a 10 MW machine with a diameter of about 178 m and a hub height of 119 m, belonging to turbine class $I_A$ according to IEC Standards (IEC 61400-1 Ed.3., 2004). The minimum, $\Omega_{min}$, and rated $\Omega_r$ rotor speed are respectively equal to 6 and 9.6 RPM. Cut-in $V_{in}$, rated $V_r$ and cut-out $V_{out}$ speed are respectively 4, 11.4 and 25 m/s, whereas reference speed $V_{ref}$ is equal to 50 m/s.

The INNWIND.EU model was implemented in the aeroservoelastic multibody code `Cp-Lambda` (Code for Performance, Loads, Aeroelasticity by Multi-Body Dynamics Analyis). Tower, blades, shaft and drive train are modeled with geometrically exact beam model (see Bauchau (2011)), whose structural sectional properties can be given as full $6 \times 6$ stiffness matrices. Aerodynamics is rendered via the classical BEM theory with hub- and tip-losses and tower shadow. First and second order dynamical models are employed to include respectively generator and pitch actuator dynamics. Tower base is linked to compliant foundation replicated through concentrated spring and dampers of suitable characteristics.

The model has been linked to the *POLI-Wind Supervisor*, an in-house `C++` software, which manage the control of the machine throughout all operational states (e.g start-up, normal shut-down, emergency shut-down, parking, idling, ...). This is a `C++` software compiled as a DLL and linked to the wind turbine aero-servo-hydro-elastic simulator, throughout the standard GH-Bladed style interface. *POLI-Wind Supervisor* is also able to call other external DLL, implementing the wind turbine controller in the power production state. The trimmer employed in this work is the open-source controller, named IK4-controller,





developed within the EU project CL-Windcon (November 2016 – October 2019), implementing a torque/collective pitch control loop and a drive train damper (IK4 Research Alliance, 2016).

The *POLI-Wind Supervisor*, thanks to its capability to manage the transition between the various operating states and to model several faults (such as grid loss, pitch actuator seizing, short-circuit, etc.), allows to evaluate the loads of the wind
turbine, subject to full list of DLC including faults and extreme events compliant with IEC 61400 Standards.

### 3.2 Optimization framework through `Cp-Max`

`Cp-Max` has been developed by Politecnico di Milano and the Technische Universität München as a tool for the integrated design of wind turbines and is continuously updated to support novel features and capabilities. A detailed description of the algorithm is provided by Sartori (2019) and by Bortolotti et al. (2016), together with a range of design applications focusing
on the development of next-generation wind turbines.

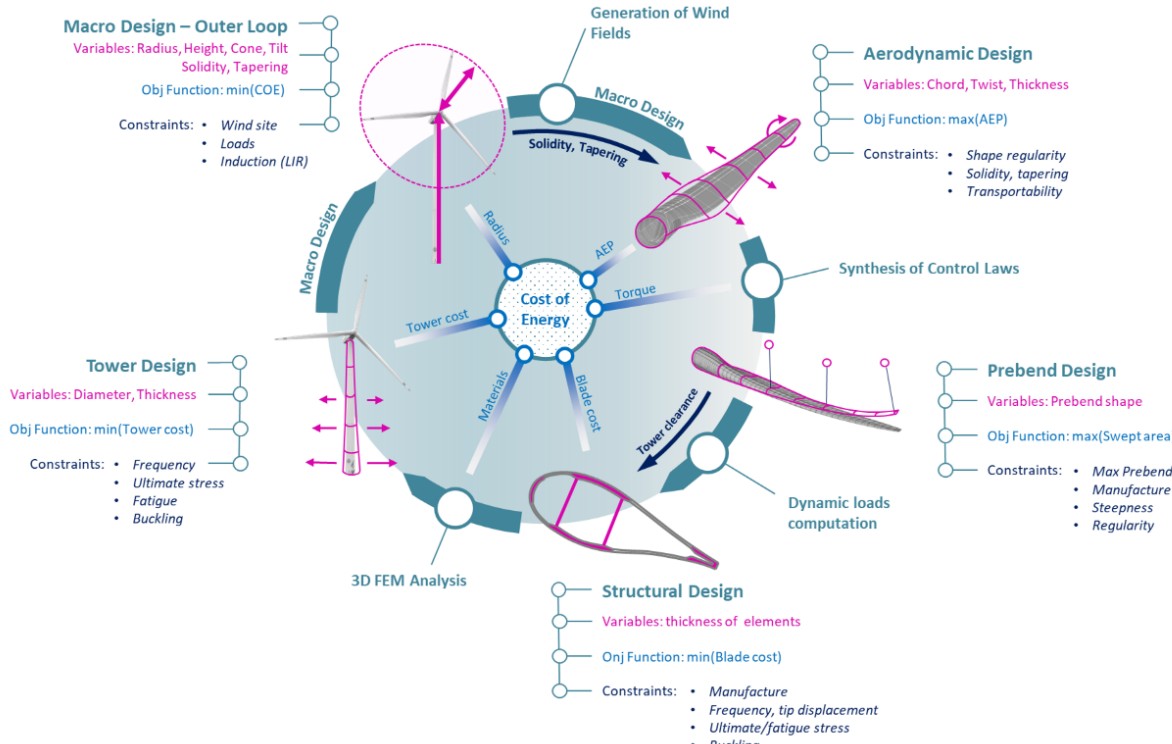

**Figure 2.** Architecture of the design tool `Cp-Max`.

The architecture of `Cp-Max` is organized into the multi-level workflow illustrated in Fig. 2. In this framework, the complete design of the wind turbine is managed by two different optimization layers: the outermost layer is the Macro Design, whose task is to optimally design a few global features of the turbine in order to minimize the COE. At this level, the design variables are the rotor diameter, the hub height, cone and tilt angles and a set of shape parameters which control the underlying design





layers. Then, for each perturbation of the Macro design variables, several submodules perform the detailed design of individual components. It must be noticed that each of the submodules is an optimization problem itself with dedicated merit function, variables and constraints. At present, several submodules are available: the Aerodynamic Design performs the optimization of the rotor shape in order to maximize the Annual Energy Production (AEP), the Prebend Design optimizes the native prebend of the blades while the Structural and Tower Design submodules conduct, respectively, the structural optimization of the rotor

and the tower to minimize the Initial Capital Cost (ICC) of the wind turbine. It is important to notice that each Submodule can either be used as a standalone tool or as part of a global design process. When both the Macro Design and the Submodules are active, the two layers continuously share informations and interface to one another in order to conduct a feasible optimization.

In this work, we use primarily the Structural Design Submodule (SDS), whose workflow is shown in Fig. 3.

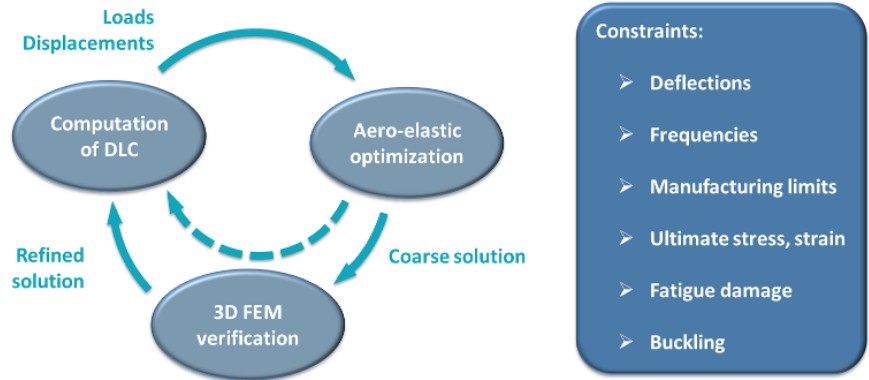

**Figure 3.** Architecture of the Structural Design Submodule.

The purpose of the SDS is to manage the structural design of the rotor through a dedicated optimization in which all the thickness of the structural components are optimized to minimize the ICC. It must be noticed that, in the current implementation of the cost model, this figure is broadly dependent on the blade total mass. As shown in the Figure, the structural optimization is conducted through a multi-steps procedure: initially, an arbitrarily large set of Design Load Cases (DLC) is computed with the multi-body dynamic solver `Cp-Lambda` (Bottasso and Croce, 2009–2018). This way, it is possible to extract the driving loads

and displacements of the turbine components from fully-resolved aeroelastic simulations in which the turbine is controlled according to the selected control strategy. Once loads and displacements are computed, the proper structural optimization is performed. At first, simplified 1D/2D models for the blade and the sections are used to achieve a *coarse* solution. Then, it is possible to add a finer verification of the structural integrity through an additional design phase in which a full 3D Finite Elements (FEM) model of the blade is analyzed. This second step, although not always necessary, allows to achieve

a better capture of localized phenomena like local stress concentration or buckling. It is important to notice that, along the whole process, structural integrity constraints are enforced according to international certification guidelines. Those account for maximum deflections, stiffness, strength, possible manufacturing limitations, fatigue and buckling.





### 3.3 Definition of Design Load Cases (DLCs) for sensitivity analysis and design process

Given the complexity of the problem at hand, any design-oriented activity should be planned out carefully as to identify the
best trade-off between modelling accuracy, computational effort and design scope. To do so, we introduced some assumptions in our redesign activity in order to make sure that the relevant aspects of this Task are thoroughly investigated and highlighted. In this view, our first choice was to limit the redesign effort to the optimization of the blade structure. In fact, although a certain impact of WF controllers could be expected on other aspects of the design, it is reasonable to foresee that an effect of such control strategies is detrimental in terms of driving loads, with important implications on the rotor structure. As a consequence,
the macro parameters of the wind turbine like the rotor radius and tower height have not been modified through this activity. Similarly, the aerodynamic shape of the rotor has been kept constant and coherent to that of the baseline. It must be noticed that, by introducing this scope limitation, it was possible to include a large set of fully-resolved DLC directly in the design, without the need to adopt any simplification on the load spectra. In particular, it was possible to use the set of DLC listed in Table 1, defined according to Standards IEC 61400-1 Ed.3. (2004). The entire set includes about 130 load cases, for a total
computational time of 26~28 hours on a common desktop.

**Table 1.** Definition of the DLC considered in the redesign.

| DLC | Wind Type | Wind speed | Horizontal Misalignment | Fault | Safety Factor | Performance indicator |
|-----|-----------|------------|-------------------------|-------|---------------|-----------------------|
| 1.1 | NTM | $V_{\mathrm{in}} : V_{\mathrm{out}}$ | - | - | 1.0 | AEP, ADC, Fatigue |
| 1.2 | NTM | $V_{\mathrm{in}} : V_{\mathrm{out}}$ | - | - | 1.35 | Ultimate |
| 1.3 | ETM | $V_{\mathrm{in}} : V_{\mathrm{out}}$ | - | - | 1.35 | Ultimate |
| 1.4 | ECD | $V_{\mathrm{r}}, V_{\mathrm{r}} \pm 2, V_{\mathrm{out}}$ | - | | 1.35 | Ultimate |
| 1.5 | EWS | $V_{\mathrm{r}}, V_{\mathrm{r}} \pm 2, V_{\mathrm{out}}$ | - | | 1.35 | Ultimate |
| 2.1 | NTM | $V_{\mathrm{in}} : V_{\mathrm{out}}$ | - | Grid Loss | 1.35 | Ultimate |
| 2.2 PF | NTM | $V_{\mathrm{in}} : V_{\mathrm{out}}$ | - | Pitch Freeze | 1.35 | Ultimate |
| 2.2 PR | NTM | $V_{\mathrm{in}} : V_{\mathrm{out}}$ | - | Pitch Runaway | 1.35 | Ultimate |
| 2.3 | EOG | $V_{\mathrm{r}}, V_{\mathrm{out}}$ | - | Grid Loss | 1.1 | Ultimate |
| 6.1 | EWM | $V_{\mathrm{ref}}$ | $-8 : 8$ deg | - | 1.35 | Ultimate |
| 6.2 | EWM | $V_{\mathrm{ref}}$ | $-180 : 180$ deg | Grid Loss | 1.1 | Ultimate |
| 6.3 | EWM | $V_{\mathrm{ref}}$ | $-20 : 20$ deg | - | 1.1 | Ultimate |

## 4 Sensitivity analysis about the effects of wind farm control on turbine level

When it comes to evaluating ultimate loads, a consideration is proper. The Standards require to design a turbine under a full list of design loads cases including normal operative conditions, situations in which the machine undergoes shutdowns possibly





concomitant with extreme events or faults, and cases in which the turbine is parked. Clearly, according to the specific case, we can either reasonably consider or utterly exclude that the turbine may operate in a wind farm controlled regime.

Imagine that for a specific reference turbine and for a specific component, the related ultimate load, noted $U^*$, comes from a case in which the wind farm control is not active (e.g. the DLC of the 6th series). It is possible and foreseeable that the wind farm control entails a general increase of machine loading, but if such increase is not enough to exceed $U^*$, the wind farm control has not any effect on that load.

Along with the list of DLCs, it is necessary to select the range in which the wind farm control is active. Clearly, the activation of the wind farm control is based on the specific implementation of the control scheme, and may depend on wind speed, turbulence intensity, geometry of the farm and even on combinations of the previous factors. In this very complex scenario, in order to simplify the analysis and make it of general validity, the farm control is considered active only in a range of wind speed (i.e. up to a given speed), no matter of the turbulence intensity or other factors.

## 4.1 Evaluation of the impact of wake redirection technique

In this Section the effects of the wake redirection control on fatigue and ultimate loads are investigated.

The wind farm control based on the wake redirection technique consists in yawing an upstream turbine by a specific amount in order to deflect its wake out of one or more downstream turbine. Within such a wind farm control scenario, if neatly performed, while the upstream machine experiences a loss of power, due to misalignment, the downstream ones produce more power thanks to a reduced wake impingement.

The turbine misalignment is reproduced in the simulations by rotating the nacelle. Positive angles are associated to counterclockwise rotations of the nacelle viewed from above. Hence, the turbine experiences positive yaw misalignment angles when the wind is coming from the right side. In the reference configuration (i.e. for a null yaw angle) the wind is assumed to blow from North to South and, accordingly, the nacelle is oriented towards North.

In order to have an investigation of general validity, without being limited to a specific turbine and a specific wind farm, a sensitivity analysis has been carried out. In particular, the effects of different steady yaw misalignment between -30 and 30 degrees on turbine fatigue and ultimate loads have been evaluated.

The 10 MW INNWIND.EU model, implemented through the software `Cp-Lambda`, was subjected to the full list of DLCs described in Tab. 1. As already said in the 1, DLC 6.x series was simulated only for the reference turbine (i.e. without misalignment), whereas DLC 1.x and 2.x for reference and for other 4 different yaw misalignment angles ($\pm15$, $\pm30$ deg).

The plot on the left of Fig. 4 shows the blade root flapwise DEL increment associated to the wake redirection for different yaw angles. These DEL loads have been computed in power production with turbulent wind from the cut-in to the cut-out wind speed (DLC1.2) and then weighted with the Weibull probability function. It's important to stress here that only for wind speed lower than 15m/s the wind farm controller (i.e. the wake steering) is active, so that at higher wind speeds there is no difference between the baseline and the wind-farm-controlled one. The same figure hence shows that the overall effect of the yaw misalignment on the cumulated DEL is limited, with an increase of slightly more than 3% at $-15\,\mathrm{deg}$. The reduction





experienced in most of the yaw range is due to the fact that in a misaligned configuration the flow velocity perpendicular to the rotor disk is lower, and entails, especially in the below-rated region, in turn lower loads on the machine.

Evaluating the impact on fatigue in the tower requires a special attention. Since the tower orientation remains fixed regardless

to the yaw angles, a separate analysis for fore-aft and side-side directions lacks in generality. In fact, one may expect the yaw angle to differently impact on the moments computed about two orthogonal directions like, for instance, fore-aft and side-side. As a consequence, the analysis must consider fatigue loads coming from the worst possible direction and, to do so, it is possible to proceed as follows.

Let define $\rho \in [0, \pi]$ a generic angle computed with respect to North and positive counterclockwise. Then, $M_\rho(t; \rho)$ is the

time-varying physical moment about a direction which is rotated of an angle $\rho$. From the time history of $M_\rho(t; \rho)$ it is possible to compute the *directional* DEL indicated with $M_{\rho\text{DEL}}(\rho)$ and to identify its highest value, $M_{\rho\text{DEL}}^{\max} = \max(M_{\rho\text{DEL}}(\rho))$ and the associated *worst direction* $\rho_{\max_{\text{DEL}}} = \arg(\max(M_{\rho\text{DEL}}(\rho)))$.

The plot on the left of Fig. 5 gives an example of the directional DEL $M_{\rho\text{DEL}}$ computed at tower base for three different yaw angles. The distributions of directional DEL has been obtained by collecting output from different load sensors at the tower

base (rotated every 15 degrees) and then interpolated. For each distribution, a diamond marker identifies the maximum load $M_{\rho\text{DEL}}^{\max}$ and the corresponding worst direction $\rho_{\max_{\text{DEL}}}$. It can be seen that for a zero yaw (solid blue line) the worst direction lies close to 90 degrees which, in the considered configuration, corresponds to both a North/South and the fore-aft bending moments. Once the rotor is rotated (dashed and dotted lines), the two directions are no longer aligned and the worst direction occurs at different angles $\rho$ depending on the yaw angle, as shown in the right plot of Fig. 5.

The plot on the right of Fig. 4 shows the increment in the maximum directional DEL $M_{\rho\text{DEL}}^{\max}$ for tower base as function of the yaw angle. Again, operating in yawed condition (for wind speed below 15m/s) does not seem critical in terms of fatigue for the tower base.

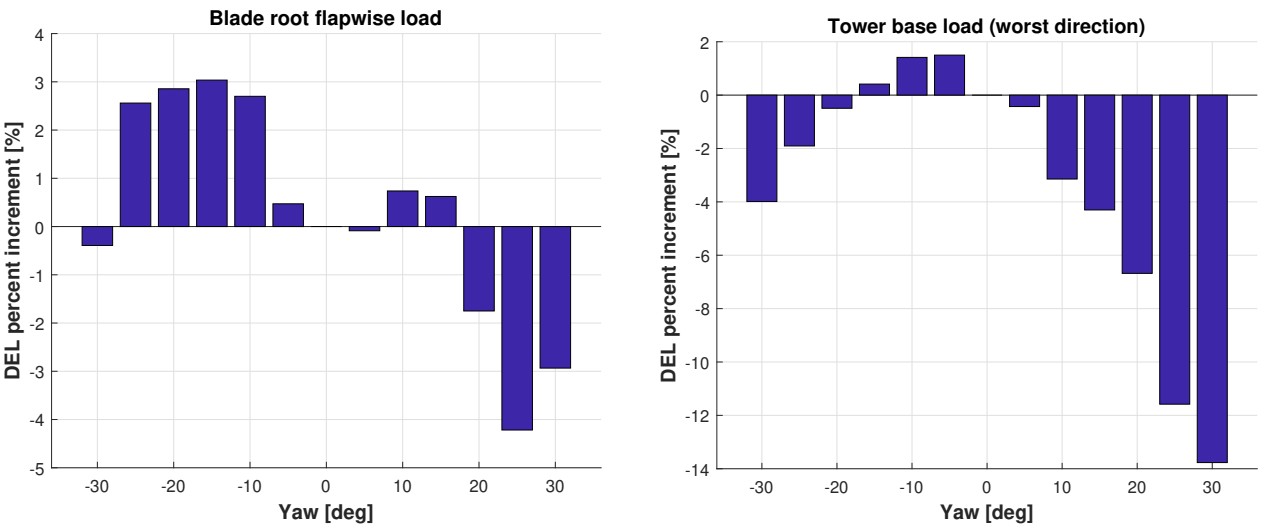

**Figure 4.** Comparison of blade root flapwise DEL (left) and tower base maximum directional DEL, $M_{\rho\text{DEL}}^{\max}$ (right).





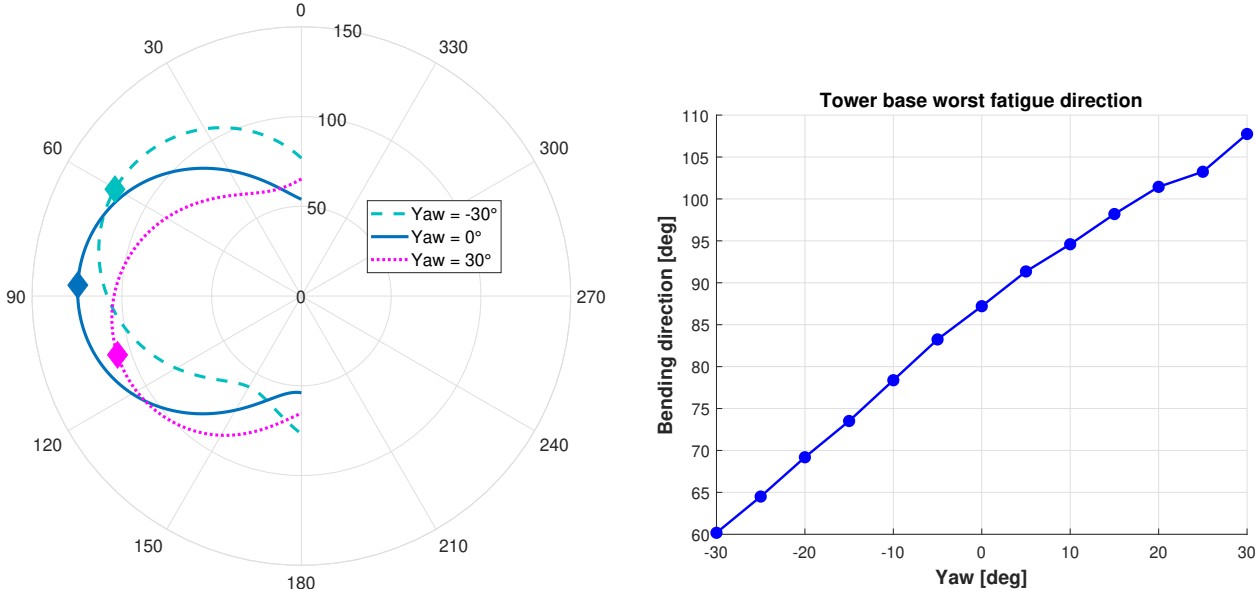

**Figure 5.** Directional fatigue at tower base. Left plot: directional DEL [MNm] at tower base for three yaw angles; diamond markers identify the maximum value. Right plot: worst fatigue direction at tower base as a function of yaw angle.

Figure 6 shows the ultimate load increment associated to the wake redirection for different yaw angles for blade root combined moment (left) and tower base combined moment (right). The text above each bar indicates the DLC which has generated such maximum loads. In terms of blade, the effect of yaw misalignment is limited with just a small increase of about 1% at 30 deg. Loading on tower seems to suffer a bit more at high misalignment with and increase of 7.5% at 30 deg.

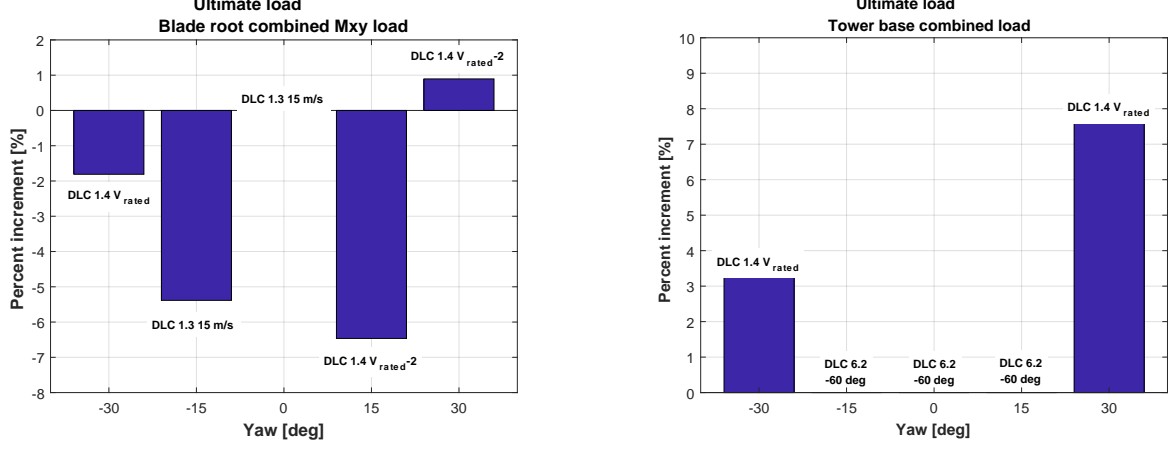

**Figure 6.** Comparison of ultimate loads. Left plot: blade root flapwise moment. Right plot: Tower base N-S moment.



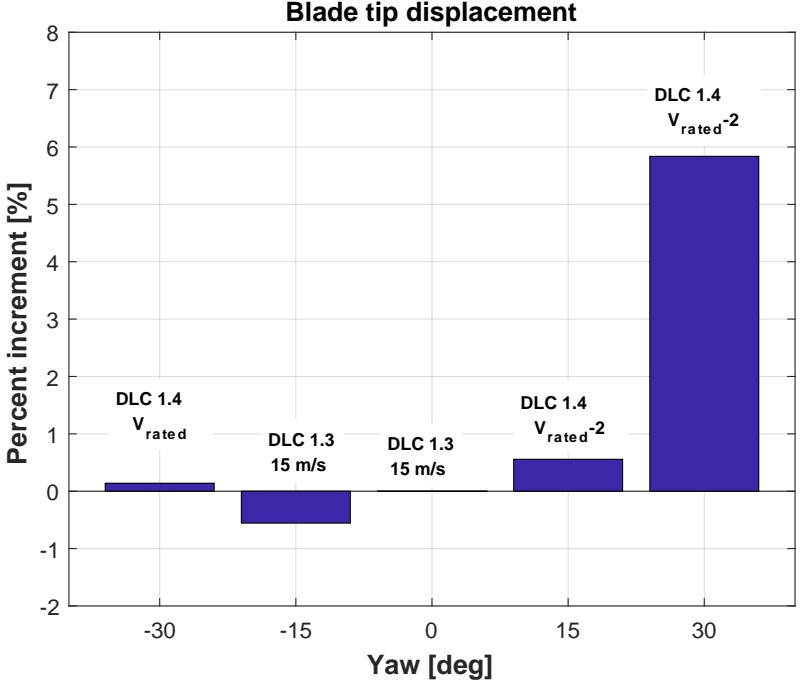

**Figure 7.** Analysis for maximum blade tip displacement

.

The analysis of blade tip displacement, on the other hand, shows an increase of tip displacement up to 6% as shown in Fig. 7. This increment, although apparently small, deserves a special attention. In fact, often, the blade design is constrained by the maximum tip deflection which is to be bounded in order to avoid dramatic blade-tower collision. Since the maximum

tip deflection typically enters in the design process as a constraint, it is possible that even a small increment of this value may lead to a violation of this constrain and in turn to the need of a blade redesign.

By looking at these results, one may also conclude that avoiding to operate at very high misalignment (i.e. at 30 deg) may be beneficial. In fact, up to ±15 deg, the increment of both fatigue and ultimate loads, seem essentially limited. It's important to stress that the approach showed here with this sensitivity analysis allows the designer to estimate the extreme parameters of

this wind farm control technology to be applied to already existing wind farm. These limits are defined by the design loads of the single wind turbine: the wind farm operators may applied this farm controller as far as the loads induced by this technology does not overtake the design ones.

In terms of actuator duty cycle, the wake redirection does not seem problematic. In fact, when a turbine operates in yawed condition, it experience lower winds. This makes the turbine work in below-rated region, with limited pitch activity, for a

higher amount of time. A quantitative evaluation shows a decrease of 41% in ADC for a misalignment of 30 deg.





Finally, AEP results to be highly affected as single machine may experience a loss up 18% at -30 $\deg$. This result is already known to the scientific community and is generally balanced by the increase in power of the downwind wind turbine(s).

## 4.2 Evaluation of the impact of dynamic induction control

### 4.2.1 Review of dynamic induction control

Another interesting means of increasing the total wind farm power consists in the so-called Dynamic Induction Control (DIC). Specifically, the upstream wind turbine, when its wake impinges on a downstream machine, modulates the thrust in a periodic way. The modulation can be performed by pitching collectively or cyclically the blades at a specific frequency or by changing the rotor speed. Clearly, the most effective action is to enforce a Periodic Collective Motion (PCM) of the blade pitches. The effect of the PCM is to dynamically vary the induction of the rotor and, hence, to increase the mixing level inside the wake.

The wake itself results to be energized by such fluctuating induction and recovers in a faster way. This techniques was recently studied through CFD simulations in two articles, Munters and Meyers (2017) and Munters and Meyers (2018) and validated via wind tunnel experimentation Frederik et al. (2019).

The DIC technique studied here is a a pure PCM at a single frequency, as described by

$$\beta_{\mathrm{PCM}} = A_{\mathrm{PCM}} \sin\left(2\pi f_{\mathrm{PCM}} t + \varphi_{\mathrm{PCM}}\right), \tag{1}$$

where $\beta_{\mathrm{PCM}}$ is the pitch setting imposed by PCM to be summed up to the pitch of the trimmer, $A_{\mathrm{PCM}}$ is the related amplitude, $f_{\mathrm{PCM}}$ the frequency, $t$ the time and $\varphi_{\mathrm{PCM}}$ the possible phase.

Albeit the limited number of studies devoted to PCM, especially if compared with the amount of literature available about wake redirection technique, it is already possible to highlight some important concepts:

– PCM seems effective in increasing the total wind farm power output by some percent, as demonstrated by both simula-
tions and wind tunnel experimentation.

– The increase in wind farm power depends strongly on the amplitude and frequency of the rotor thrust variation.

– Rather than in terms of frequency $f$, the effect of the PCM technique is to be viewed in terms of the dimensionless Strouhal number $S_t$, defined as

$$S_t = \frac{f_{\mathrm{PCM}} U_\infty}{D}, \tag{2}$$

being $U_\infty$ the undisturbed wind velocity and $D$ the rotor diameter.

– The optimal Strouhal number was found to be 0.25 in CFD simulationMunters and Meyers (2018), whereas in wind tunnel it was possible to verify significant power increases in a wider rage between 0.17 and 0.45Frederik et al. (2019).





### 4.2.2 Effect of amplitude and Strouhal number on turbine loading

Different couples of amplitude $\beta_{\mathrm{PCM}}$ and Strouhal number $S_t$ were considered: the range in amplitude was set between 1 and
4 deg, whereas the range of Strouhal between 0.2 and 0.5.

At first, consider AEP, ADC and fatigue loads (DLC 1.1 and DLC 1.2), again under the assumption that PCM is active
only below 15 m/s. The loss of AEP of the upstream wind turbine results very low. As example, the case of $\beta_{\mathrm{PCM}} = 2$ deg
and $S_t = 0.5$ is associate to a loss of AEP equal to 0.5%. On the other hand, considering an higher amplitude, $\beta_{\mathrm{PCM}} = 4$, the
decrease of AEP results 2.2%.

Dealing with Actuator Duty Cycle (ADC), the effect of PCM is way more significant. In particular, considering $S_t = 0.5$,
the increase of ADC results equal to 77% and 143% respectively for $\beta_{\mathrm{PCM}} = 2$ and $\beta_{\mathrm{PCM}} = 4$.

Figure 8 shows DEL as a function of the wind speed for blade root flapwise moment (left) and tower base N-S moment
(right), in the case of $\beta_{\mathrm{PCM}} = 2$ and $S_t = 0.5$, as an excerpt from the results. The increase of DEL of the PCM (dashed line)
with respect to the reference, i.e. without PCM, turbine (solid line) is pretty visible. Apparently, higher velocities are associated
to more significant increases. This fact demonstrates that the maximum wind speed at which the farm control is active is to be
carefully selected, in order to find a good balance between optimization of power output and increase of loading.

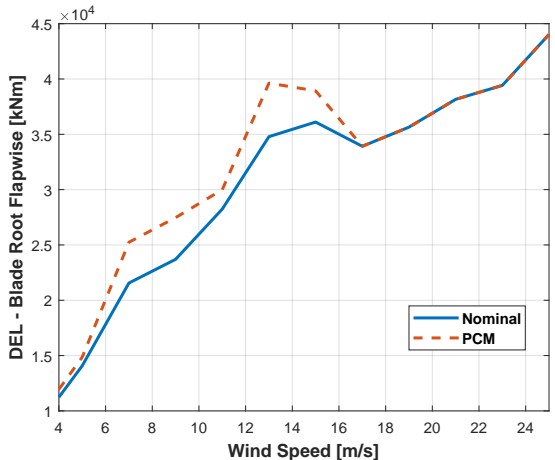 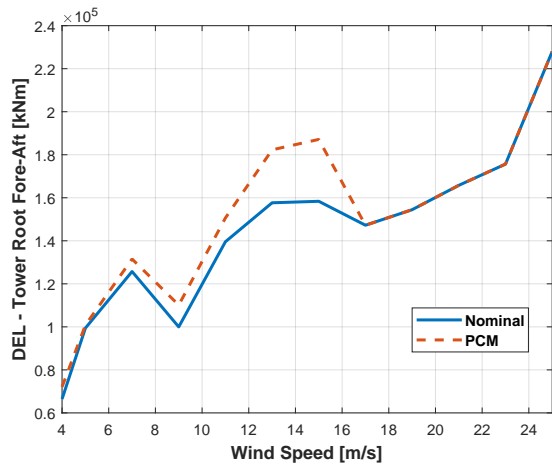

**Figure 8.** Comparison of DEL. Left plot: blade root flapwise moment. Right plot: tower base N-S moment.

A sensitivity analysis was also performed so as to compute the effect of PCM in term of DEL as functions of Stroual number
and amplitude. Figure 9 shows the percentage DEL increment for different combination of Strouhal number and amplitude
for blade root flapwise (left) and tower base (right). As expected, the highest increases are associated to larger ampliture and
Strouhal.



For both blade flapwise and tower base, the effects are significant and may arrive to the 20% and 30% respectively in the worst cases ($\beta_{\mathrm{PCM}} = 4$ and $S_t = 0.5$). However, if one excludes the highest amplitude limiting the authority of PCM to 2 deg, the detrimental effect of PCM in terms of fatigue results acceptable being equal to about 10% for both loads.

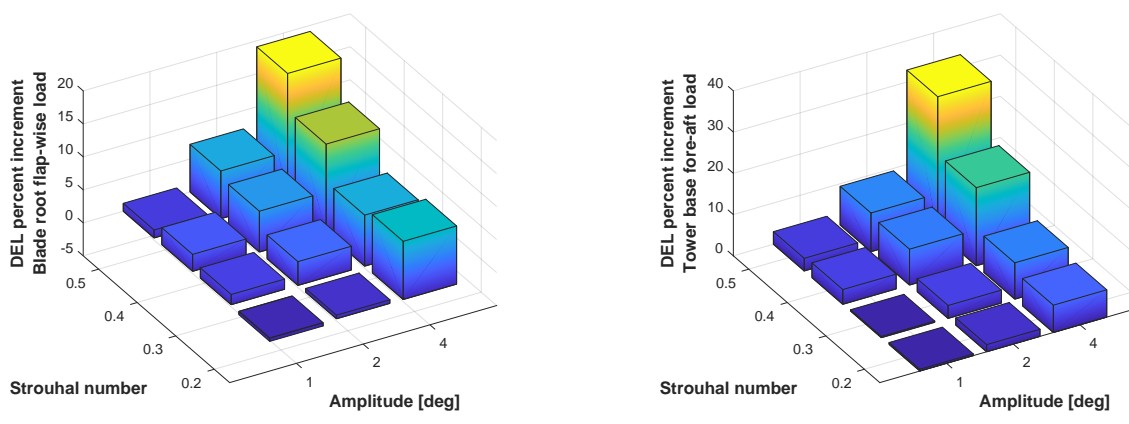

**Figure 9.** DEL increase as function of amplitude and Strouhal number. Left plot: blade root flapwise. Right plot: tower base N-S moments .

The DEL on the blade root edgewise, not shown here for the sake of brevity, is mildly influenced (increase of about 1.5% in the worst case). This results is not surprising as the loading in edgewise direction is mostly governed by gravity rather than aerodynamics. Similar conclusions can be derived for nodding, yawing and tower top moment, depicted in Fig. 10, for which the effect of PCM is quantifiable in less than 1.5%.

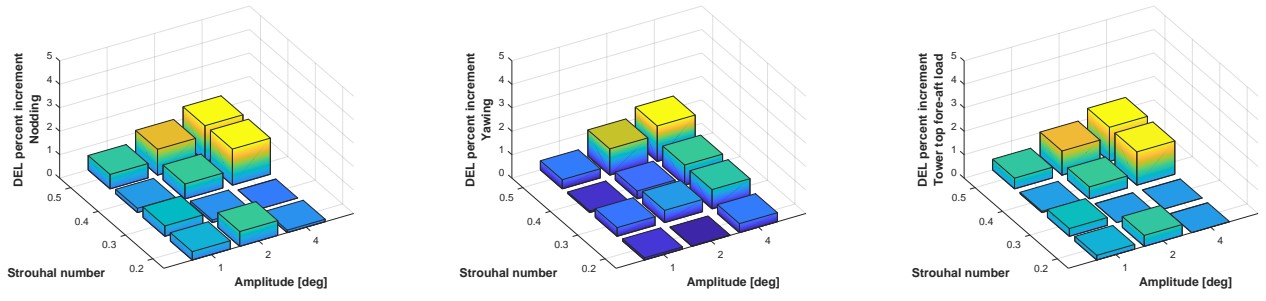

**Figure 10.** DEL increase as a function of PCM amplitude and Strouhal number. Left plot: nodding moment. Center plot: yawing moment. Right plot: Tower top N-S moment.

A detailed analysis concerning the ultimate loads has been carried out as well. It was primarily observed that the phase of the oscillation $\varphi_{\mathrm{PCM}}$ is of paramount importance. Consider for example the set of DLC including extreme conditions. By chance, it is possible that the particular extreme event, like a gust or a fault, occurs at a time in which the PCM control is increasing the collective pitch reducing the rotor loading. Clearly, in this situation, the peak load involved by the extreme event will be




probably smoothed out and one could be erroneously led to believe that the control has a beneficial effect. Conversely, if the extreme event had occurred in correspondence of a decrease of the collective pitch, the effect of the control would have been the one of increasing the peak load.

In order to find the worst case, i.e. that condition which maximizes the increase of the peak load, 8 different 45-deg-spaced phases have been considered. Consequently, the full set of DLC in Table 1 were simulated for each couple amplitude-Strouhal 8 times by varying the phases of the oscillation.

    Figure 11 shows the percentage increment for different combinations of Strouhal number and amplitude for blade root combined moment (up left) and tower base combined moment (up right) and hub combined moment (down). Above each bar,

a text indicates the DLC which has generated that maximum load. Significant increases are only associated to blade loads, whereas hub and tower result to be not affected by PCM.

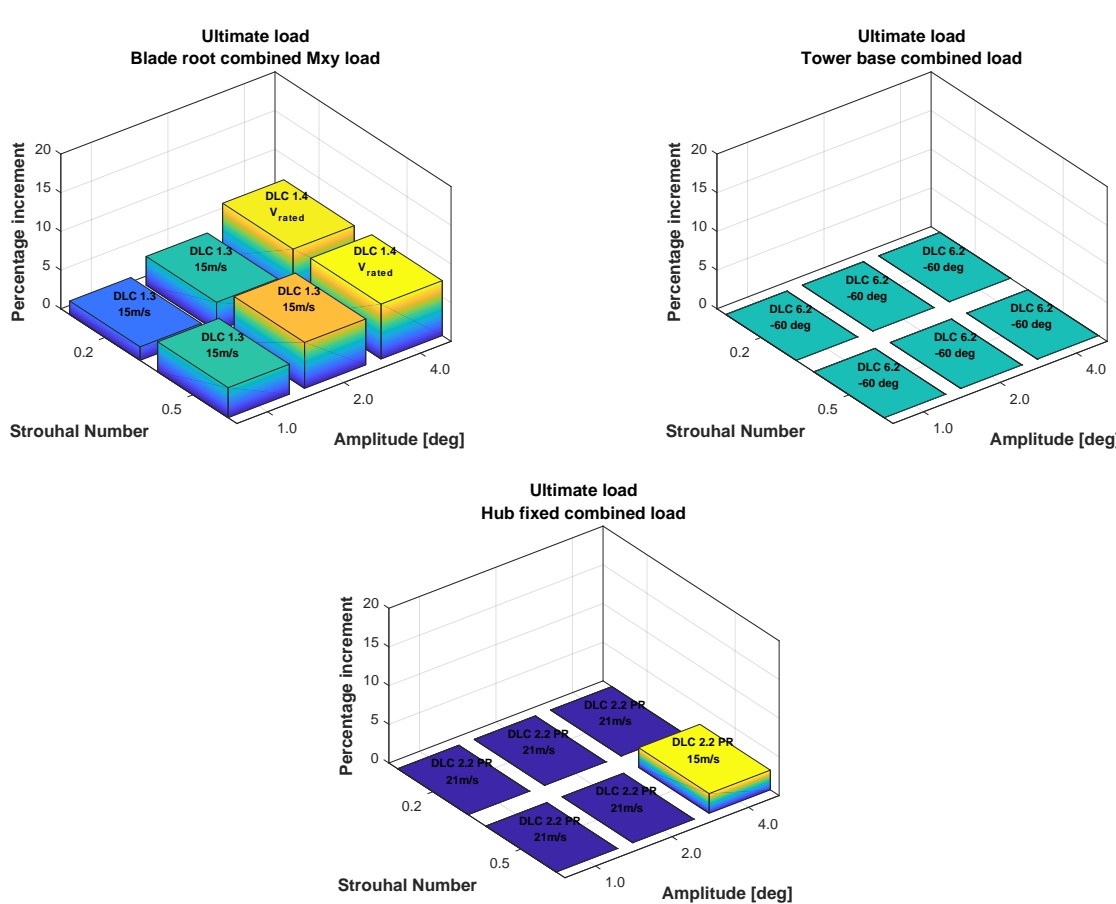

**Figure 11.** Ultimate loads as function of amplitude and Strouhal number. Up-left plot: blade root flapwise moment. Up-right: tower base combine moment. Bottom plot: hub combined moment.

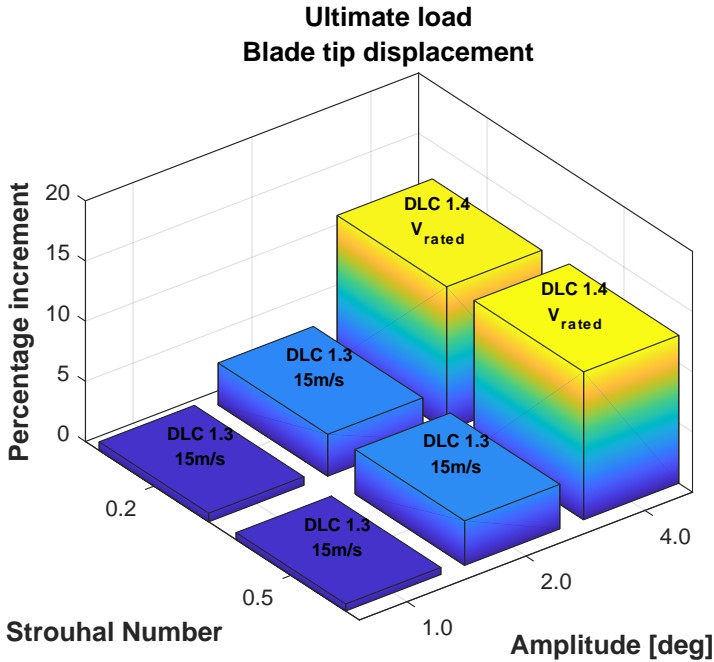

**Figure 12.** Sensitivity analysis for maximum blade tip displacement

.

A sensitivity analysis was also conducted so as to compute the variation of maximum blade tip displacement. An increase up to $12\%$ is measured for $\beta_{PCM} = 4$ deg, as shown in Fig. 12. This indication is extremely important as the maximum tip deflection typically enters as an active constraint into the design of the blade, affecting the thickness of its structural elements.

If one exclude the highest PCM amplitude (i.e. 4 deg), the maximum tip deflection increases only of 3% which may correspond to a lower impact on blade design.

### 4.3    Comparison between periodic collective motion and wake redirection for the 10 MW INNWIND.EU turbine

Table 2 shows a comparison between the worst cases of WR control and PCM for the 10 MW INNWIND.EU turbine.

At first sight, PCM control with amplitude of 4 deg appears to be extremely impacting in terms of both fatigue and ultimate

loads, with an increase of more than 10% in the maximum tip deflection which could be considered excessive, at least for the present machine whose design is constrained by this value.

If one considers, on the other hand, only the PCM with amplitude of 2 deg, the impact of PCM and WR becomes comparable, even though the latter seems less impacting especially for fatigue. In terms of ultimate loads, PCM has an higher impact on blade loads and null on tower, while the opposite happens for blade redirection.





**Table 2.** Comparison of worst cases (PCM vs WR)

| | PCM | | WR |
| --- | --- | --- | --- |
| | $A_{\text{PCM}} = 2$ deg | $A_{\text{PCM}} = 4$ deg | |
| AEP | -0.5 % | -2.2% | -18.4 % (at 30 deg) |
| ADC | +77 % | +143 % | -41 % (at 30 deg) |
| DEL blade flap | +7.2 % | +19.9 % | +3.5 % (at 30 deg) |
| DEL tower base fore-aft | +9.4 % | +32.9 % | +0.7 % (at 30 deg) |
| DEL hub | +1 % | +1 % | $\approx$ 0 % (at 30 deg) |
| Ultimate blade root combined | +5.9 % | +7.1 % | +1.0 % (at 30 deg) |
| Ultimate tower base combined | 0 % | 0 % | +7.9 % (at 30 deg) |
| Ultimate hub combined | 0 % | 2.5 % | 0 % |
| Max tip deflection | 3 % | 12.3 % | 5.8 % (at 15 deg) |

Maximum tip deflection needs a special attention as both control techniques entail a significant increase in this quantity. In fact, for a typical blade design based on glass, the blade-to-tower clearance is often an active constraint of the structural design Sartori (2019); Bortolotti et al. (2019). Moreover, in this case the load case bearing the ultimate displacement is DLC 1.4 referring to an ECD, that is, a deterministic wind case. This suggests that an ECD may happen regardless of the turbulence intensity which justifies, at least for the present study, the initial choice of neglecting a dependency between the TI and the 350   activation of the wind farm controller.

## 5   Evaluation of the impact of wind farm control on rotor design

In the Sec. 4, different wind farm controls have been analyzed with the aim of computing the related effect on wind turbine fatigue and ultimate loads, as well as on other important design parameters, such as the maximum blade deflection and actuator duty cycles. The performed analyses showed that WR and PCM have a limited but not negligible effect on fatigue and, more 355   important, a significant impact on ultimate loads, especially on the maximum tip deflection, which is a typical design driver for blades (i.e. a maximum blade tip displacement is severely enforced in the design process in order to maintain a suitable clearance between blades and tower).

This Section deals with the impact of wind farm control within the design of the turbine rotor, and can be considered as the subsequent step with respect to the study of Sec. 4.

The goal of this analysis is to quantify the possible modifications on the structural design of the blade if wind farm control are considered. Possible increase in blade mass and turbine cost will be considered as concise indicators of the impact of wind farm control on blade design.

Since the focus of this study is on the macroscopic impact of WF controllers on the design rather than to provide a fully-feasible structural layout, we limited our analysis to the aero-elastic optimization loop of Fig 3.





In order to perform a neat comparison where the effects of the sole wind farm control are highlighted, all redesigned rotors should be "optimal", in the sense that they should be all coming from an equal optimal design process characterized by the very same cost function and constraints, otherwise, it would be impossible to split the effects of the WF controller from those of the specific optimization strategy in the final comparison. To this end, the reference INNWIND.EU 10 MW wind turbine is firstly redesigned with `Cp-Max` following the procedure described in Sec. 3.2, yielding the "baseline" rotor. Then, the baseline
configuration will be updated by the same optimization process but including this time the presence of the wind farm control. The considered wind farm control is the PCM characterized by $S_t = 0.5$ and $\beta_{\mathrm{PCM}} = 2$ deg.

    The design process of the baseline generated an optimal solution compliant to all optimization constraints, with a structure mildly different with respect to the nominal INNWIND.EU. Hence, for the sake of brevity, the related detailed analysis is not reported in this manuscript.

## 5.1   Structural redesign with PCM

The structural optimization was then repeated by taking into account both the standard DLC set from Table 1 and all the DLC in which the turbine is controlled with the PCM. Different phase angles of the PCM were included in the ultimate loads/displacements analysis. Once again, the entire set of DLC was re-computed at each structural iterations to make sure that, as the structural design evolves, the loads are updated accordingly.


The main results of the redesigned rotor with the PCM are summarized in Table 3. As shown, the introduction of the WF controller leads to a general worsening of all key performance indicators. It must be stressed, however, all indicators refer to the individual turbine as the current release of `Cp-Max` implements a turbine-specific cost model and a proper assessment of the impact of the PCM on the cost of energy should be evaluated at wind farm-level.

**Table 3.** Comparison between the KPIs of the Baseline 10 MW and the redesigned rotor.

| Performance | Baseline 10 MW | Redesign PCM | Variation |
|---|---|---|---|
| Blade mass: | 40643 kg | 45436 kg | +11.8% |
| AEP: | 45.86 GWh | 45.63 GWh | -0.5% |
| CoE: | 89.42 EUR/MWh | 90.22 EUR/MWh | +0.89% |

For what concerns the aerodynamic performance, Table 3 shows that only a slight deterioration of the AEP is expected when the PCM is used. This is mainly due to the collective pitch motion imposed in the partial-loading range of the power curve which results in the collective pitch of the blades being cyclically driven away from their theoretical optimum value. This is confirmed by looking at the time-averaged turbulent power curve given in Fig. 13 (left) and the corresponding power coefficient (right). However, this detrimental influence is partially compensated by the increased stiffness of the blades which
contribute to preserve the AEP by reducing the deformations experienced by the rotor. Another important feature of the PCM



is the increase of Actuator Duty Cycle (see Fig. 14 (left)) which is a direct consequence of the increased service time required to the pitch actuators.

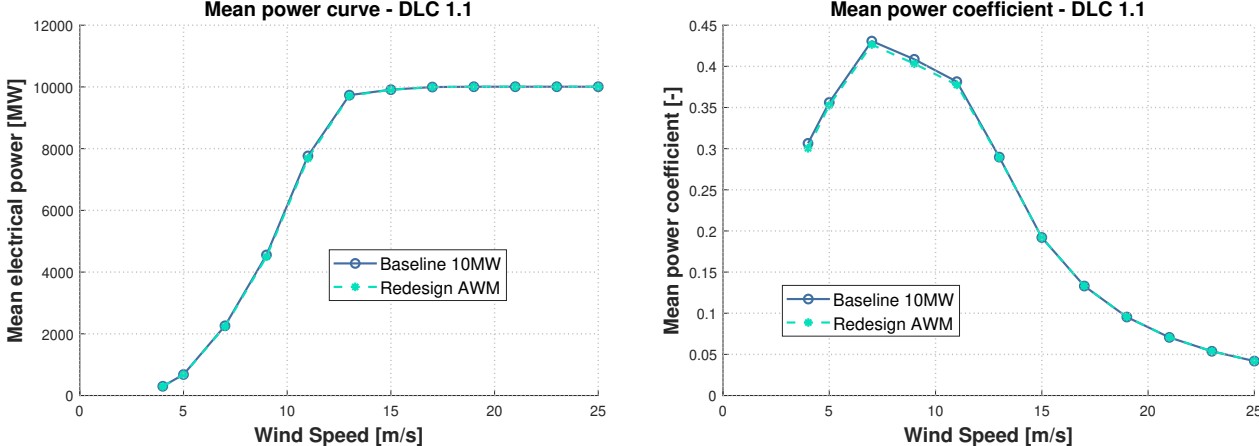

**Figure 13.** Redesign process including PCM. Left plot: time-averaged turbulent power curve. Right plot: power coefficient.

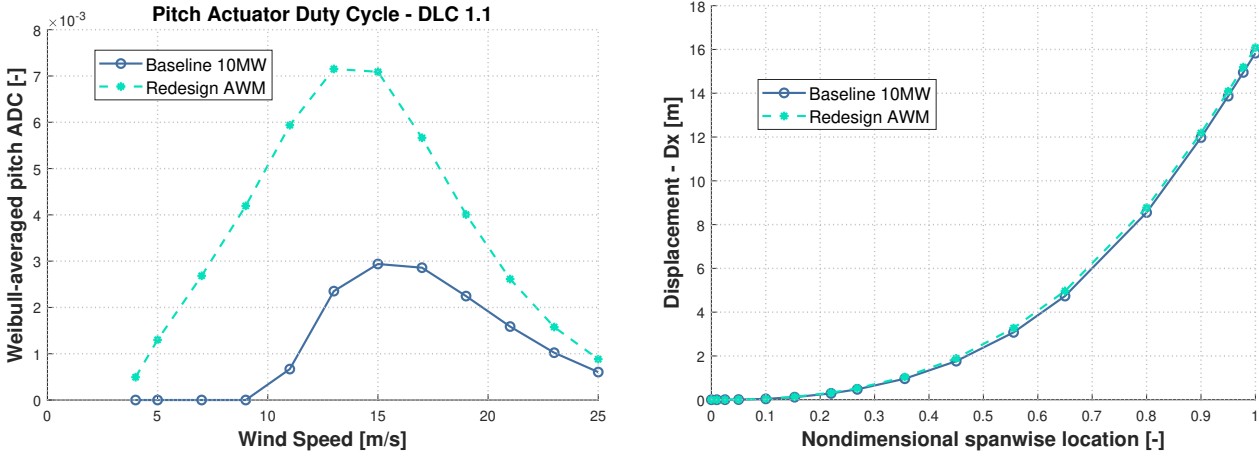

**Figure 14.** Redesign process including PCM. Left plot: Weibull-averaged ADC. Right plot: ultimate blade deflection.

From a structural perspective, as expected the introduction of the PCM resulted in an about 12% higher blade mass. This result comes from a combination of higher loads and higher displacements introduced by the controller. Specifically, the

increased blade deflection required a heavy redesign of the spar caps in order to avoid the violation of the constraint on the maximum tip displacement. The optimal distribution of spar cap thickness is shown for both the baseline and the redesigned rotor in Fig. 15. It is worth to mention that, due to the increased flapwise stiffness, at the end of the optimization the maximum displacement of the redesigned rotor is almost identical to that of the Baseline and, most importantly, it does not exceed the allowed tower clearance (see Fig. 14 (right)).



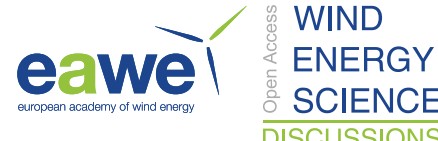

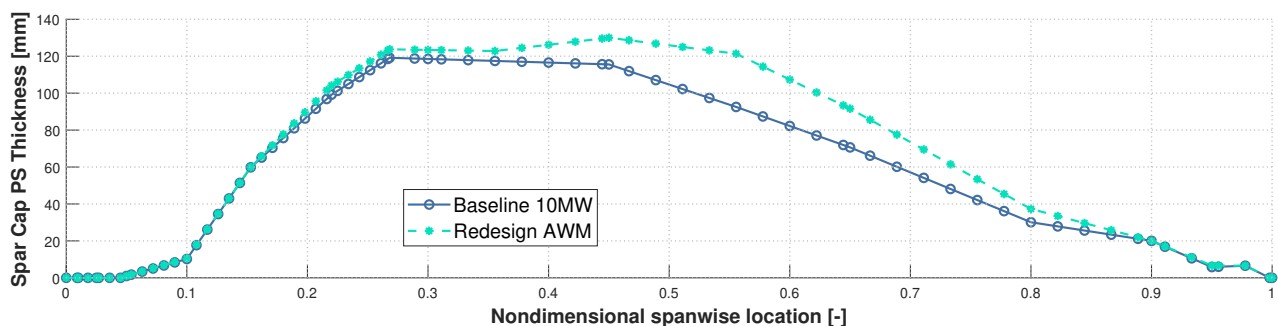

**Figure 15.** Redesign process including PCM: thickness of the spar caps.

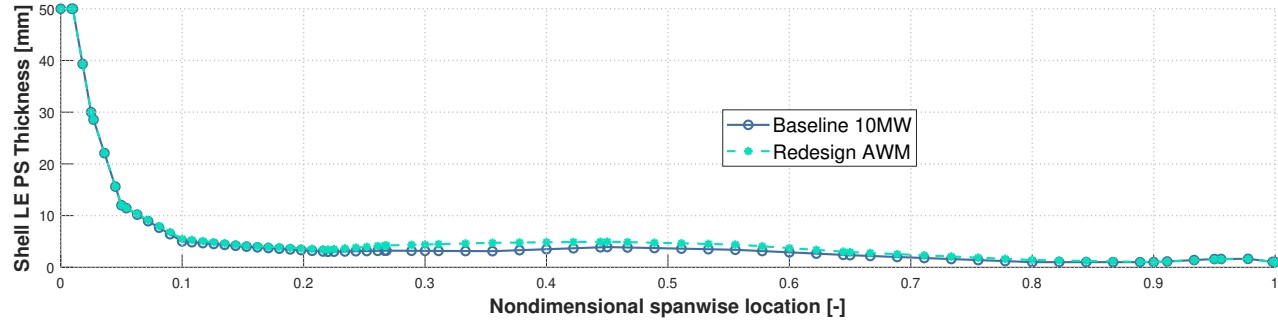

**Figure 16.** Redesign process including PCM: thickness of the shell.

The fatigue DEL show a general increase of the loading in the rotor, although the impact on the fixed infrastructure like hub and tower is less significant. This required a slight redesign of both the shell and the shear webs, as both elements are heavily driven by fatigue. This can be seen in the corresponding distributions of thickness provided by Fig. 16 and Fig. 17.

It must be noticed that part of the fatigue increase is directly ascribable to the use of the PCM controller but an important contribution is due to the increased blade mass. Then, an important conclusion is that fatigue loading should expect to increase

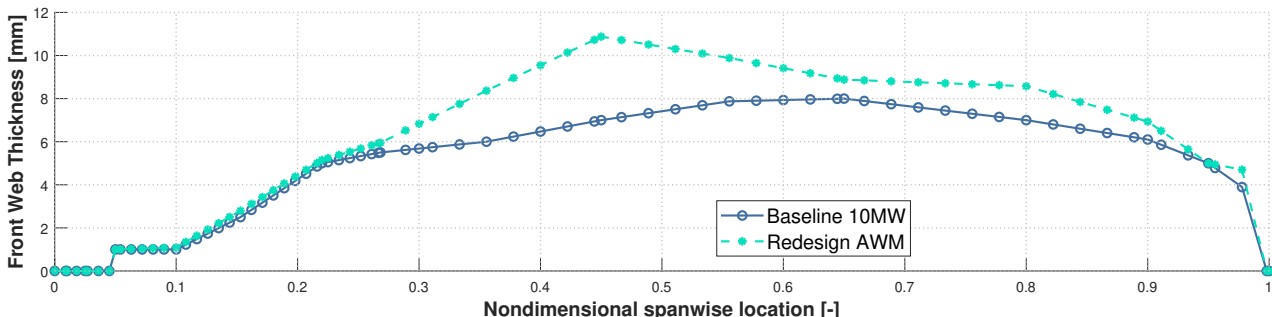

**Figure 17.** Redesign process including PCM: thickness of the shear webs.





not only for the use of the WF controller, but also due to the unavoidable mass increase required by the higher ultimate loads and displacements.

    A complete comparison of the fatigue loads of the Redesign PCM rotor against the Baseline 10MW is given in Fig. 18, in which equivalent loads are made non-dimensional with the corresponding values of the Baseline 10MW. Here, `flp` identifies flapwise, `edg` is edgewise, `trs` is torsion, `thr` is thrust, `nod` is nodding, `yaw` is yawing, FA and SS are, respectively, fore-aft

and side-side loads. As already discussed, the largest impact of the redesign in terms of fatigue is detected in the blade loads, in the hub thrust and, slightly, at the tower base fore-aft. The increase at tower base, however, would hardly justify a complete redesign of the tower.

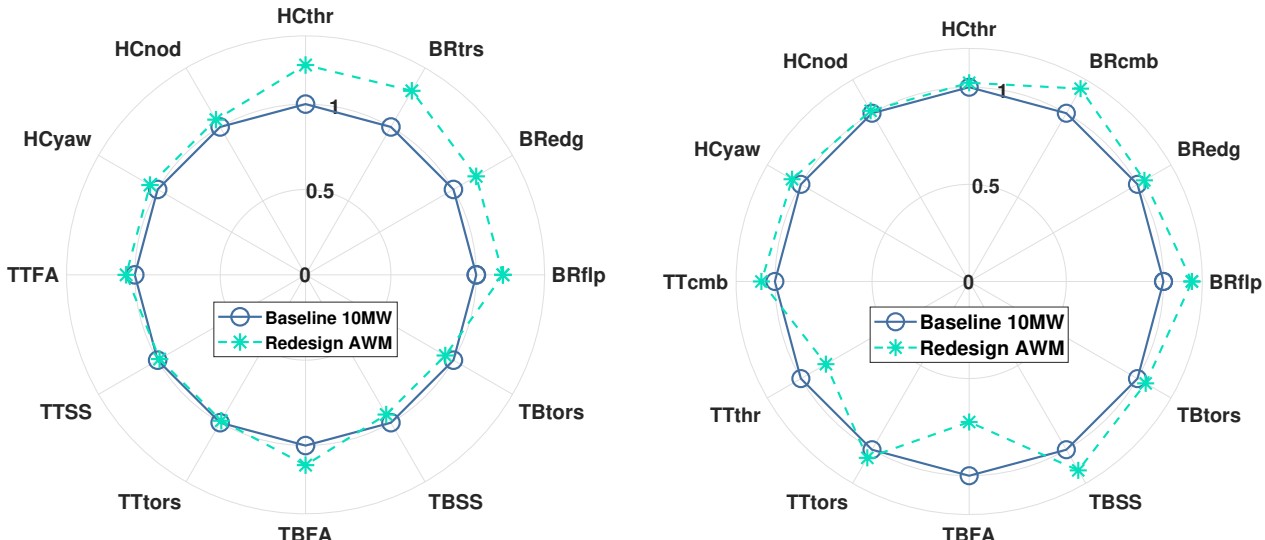

**Figure 18.** Comparison between PCM-redesigned and Baseline 10MW turbine. Left plot: fatigue DEL. Right plot: ultimate loads. 'BR' is blade root, 'HC' is hub center, 'TT' is tower top and TB is tower base

    The evolution of the ultimate loads follows a similar trend, with the worst effect being detected on the rotor. A direct comparison of the ultimate loading of the Baseline 10 MW and the Redesigned PCM is given in Fig. 18 (right). The nomenclature is

similar to that of the fatigue analysis, however, here the tag *cmb* identifies multi-directional combined loads. From the diagram it is possible to notice how the blade loads are globally increased by the combination of PCM and higher blade mass. When it comes to the fixed infrastructure, however, the impact is mixed. On one side, the side-side and the torsion at the tower base are slightly increased, while the fore-aft is significantly reduced. Based on this limited analysis, then, it is possible to conclude that the introduction of the PCM as a WF controller would require a redesign of the rotor but, likely, would not affect the structural

integrity of the hub and the tower.





# 6    Conclusion and Outlook

In this paper, a procedure to evaluate the effects of wind farm control techniques on a single wind turbine level have been developed. Two difference control types have been considered, the dynamic induction control and the wake redirection, and tested on a reference 10 MW wind turbine.

The entire study is based on the inclusion of the farm control in the current status of the Standards and is defined so as to consider the worst scenario. To this end, a possible dependency of the farm control from the turbulence intensity was intentionally neglected, allowing also for a simplification of the problem. However, the need of updating the Standards and especially through a dedicated set of design load cases, tailored for the presence of wind farm control is, at least to Authors' opinion, necessary.

The study has been performed through two steps. At first, the impact of the controls is evaluated in terms of important indicators (fatigue and ultimate loads, ADC, maximum blade tip displacement and AEP. The analysis has been conducted as a sensitivity study where all indicators are computed as functions of global parameters of the controllers themselves. The scope of this first step is twofold. First, this approach allows the operator to quantify the impact of the wind farm control algorithm on the single, already existing, wind turbine. In this way it's possible to define the operational limits of the wind farm controller,

i.e. the thresholds of the control parameters, which ensure that the design loads of the wind turbines are not exceeded. Second, this step can be viewed as preliminary for a design process of a new turbine which considers also the presence of a wind farm control. In this second step, showed in the previous section with the dynamic induction control, a complete structural redesign of the rotor has been performed, in order to quantify the difference in the structural design of the blade induced by farm control.

From the analyses performed in the present work on a 10 MW wind turbine, the following conclusion can be derived.

– The impact of both controls in terms of fatigue is significant but does not seem particularly dramatic, especially if one considers that the wind farm control could be active for a limited range of wind directions, i.e. for only those directions for which a wake impingement is detected on downstream rotor.

– In some cases, the wake redirection control entails a reduction of the fatigue. The reason for this is to be found in the fact that operating in yawed conditions reduces the normal component of the wind with respect to the rotor disk, lowering

loads.

– The most impacting control seems the one based on the dynamic induction control, especially if higher amplitudes of the blade pitches are considered. However, if the amplitude are maintained below 2 deg, the potential increase in fatigue results limited.

– Tower ultimate loads are particularly affected by wake redirection, with the ECD condition being the most impacting

DLC.

– The blade loads, on the other hand, are mostly affected by dynamic induction control.



- Both wake redirection and dynamic induction control entail an increase of the maximum tip deflection. This fact represents an issue for the blade of the reference 10 MW machine, considered in this work, as the related design is constrained by the tip displacement. However, it is possible to imagine that other rotors may be prone to the same problem as the blades are typically designed under severe maximum tip displacements enforced to maintain a suitable clearance between blade tip and tower.

- A dedicated redesign of the blade in the case of dynamic induction control has been performed. Since this control type induced a non negligible increment of the loads and of tip deflection, a general increase of the structural parts of the blade was obtained after the optimal design process.

- The new redesigned blade undergoes and increase of about the 10% of the mass, which does not seem critical for real application of such a farm control.

- Due to the increase of mass, also the fatigue loads on blades increases in the redesigned blade.

It is important to stress that the obtained results refer to the considered turbine and may be different for different machines, hence, in a possible extension of this work one should consider turbines of different sizes and classes.

Moreover, also the design of tower, being affected by the farm control as well, could be inserted in order to gain a better knowledge of the impact of wind farm control on entire turbine.

Another important aspect deals with the assessment of the economic impact of farm control at wind-farm level. To this end, specific studies should be performed so as to consider in the design process, along with the modification of turbine structure, the possible increment in the AEP of the entire farm. With the final goal of computing the difference of the cost of the energy associated to the farm control, one has to have first a tool for evaluating the power harvested by the farm in different conditions, to be weighted by wind weibull and rose and finally a cost model for wind farm.

All aforementioned points, out of the scope of this paper, are currently under investigation.

*Author contributions.* All authors provided fundamental input to this research work through discussions, feedback and analyses of the state of the art. PDF conducted the sensitivity analysis with the WR control, SC conducted the sensitivity analysis with the PCM, LS performed the structural design, AC developed the main idea behind this work and supervised the research activities. SC, AC and LS wrote the manuscript.

*Competing interests.* No competing interests are present

*Acknowledgements.* This work has been partially supported by the CL-Windcon project, which receives funding from the European Union Horizon 2020 research and innovation program under grand agreement No. 727477.



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
