# Peer review of "Evaluation of the impact of active wake control techniques on fatigue and ultimate loads for a 10 MW wind turbine."

_Wind Energy Science, 2019_

## Referee Comment (RC1) · Anonymous Referee #1 · 18 Feb 2020

This is an interesting topic, and the conducted research and associated conclusions are worthy of publications. But the clarity of the writing is quite poor. There were many sentences that i fundamentally could not understand, and most of the paper has rather awkward English. It really needs a review and edit by a native speaker. Once that is complete, i would be willing to re-review, and i think that will go smoothly since the actual research is quite interesting.

---

## Referee Comment (RC2) · Anonymous Referee #2 · 9 Mar 2020

**General comments**

This paper is in a very relevant and active research area of wind farm control. Many studies have been published in the last decate focused on the benefits of different wind farm control strategies, such as wake redirection and induction control. However, there has been only limited attention on the impact of wind farm control on the loads of individual wind turbines. These papers have not been cited well enought in the introcution section, and the actual contribution of the paper is not clearly motivated by building up on what is already known.

My most important comment, however, is the scope of the study. Fatigue loads in a wind farm cannot be evaluated on a single wind turbine level. A single turbine operating with yaw misalignment in free stream as first in a row may increase its loads

for some yaw misalignments. Behind the turbine, however, the downstream turbines might benefit reduced fatigue loading as the wake is moved away from them. Considering one single turbine in free stream does not provides completely no basis for drawing conclusions on the overall fatigue loading. Instead, all wind directions must be considered in a detailed analysis, including the proper wind speed and direction distributions. Therefore, I cannot approve the paper unless one of the following modifications are made:

- fatigue loads are completely removed and the discussion focusses on ultimate loads only (which I would recommend), or

- it is clearly stated that the fatigue load analysis performed is too limiting to draw conclusions about the overall impact on the lifetime fatigue loading on any turbine in the wind farm, and that further research is needed to study these effects. In doing so, the presented results are to be compared to earlier results by other researches. Notice that the fatigue results, as they stand now, will still not provide much added value in terms of what is already published.

The part of the paper related on ultimate loads is relevant and novel. In my opinion, this should be the focus of the paper. Also here, the story needs to be put in the right perspective. You can't just claim that because of the sensitivity type of analysis, the results and conclusions can be generalized to any wind turbine type. This is obviously not true, as extreme loads depend on many aspects, such as turbine aerodynamics properties, wind conditions, and (supervisory) control system. This needs to be mentioned.

The part on blade redesign is obscuring the focus of the paper, and given its current length, I propose to completely remove Section 5. Same holds for Sections 3.2 - 3.3 (also related to redesign), although Table 1 should remain detailing the load cases.

Furthermore, there are lots of typos and other errors, please revise the language toroughly before resubmitting.

I will be very willing to review the paper after a major revision following above recommendations.

---

## Referee Comment (RC3) · Vasilis A. Riziotis (Referee) · 27 Apr 2020

The paper assesses the effects of two particular wind farm control methods i.e. WR and DIC on the ultimate and fatigue loads of the turbines that the wind farm controller commands to take control action (usually the ones of the front rows of a wind farm). Then they re-design the rotor blades for the DIC one which proves to be the most critical, both in terms of strength and fatigue. In the reviewer opinion the paper addresses a very interesting and important topic and deserves publication after some revision is made to the original text, including the re-polishing of the language (among others some suggestions for language corrections are given in the supplement pdf). Please see below my main concerns and points to be further elaborated in the revised text: 1) By reading the title, very large expectations are created to the reader, that the

actual impact of the wind farm control on the design loads will be assessed. However, as explained in section 1 and 2, the work turns out to be a parametric study of the effect of i) yaw misalignment and ii) periodic collective pitch angle variation on design loads. Finally the re-design of the blade is only needed and performed for the latter. In order to support originality of the proposed work I would recommend the authors to try to link the conditions scanned in 4.1 and 4.2 with the actual expected conditions in the occasion of wind farm control. 2) There are several independent studies which indicate that overall, yaw misalignment, positive and negative increase the DELs of the flapwise bending moment (especially as it increases towards -/+20-30deg). In the reviewer opinion, some more convincing explanation of why this is not predicted by the present work must be given (e.g. some time series plot explaining this reduction etc.). A reference on the yaw correction model used in Cp-lambda is also missing. This is very important in order to accurately predict loads variations in yaw. Moreover, it is stated that the DELs decrease because of the reduction in the mean value of the load. It would be nice to provide the formula of the DEL calculation used by the authors, as the standard one, that the reviewer considers, does not involve the mean load value but only the ranges of load variations. Furthermore, it would be nice to provide the DEL reference frequency and exponents used in the different components DELs calculation (i.e. blades and tower). 3) With respect to DIC the results are as expected. One point that perhaps needs some more attention is to give an indication of how far these perturbations in the wake flow generated by the upstream turbines, travel. Of course they facilitate mixing in the wake but do you also expect a fast decay of the low frequency coherent fluctuations introduced by collective pitching? These may cause additional low frequency wind inflow variation to the downstream turbines which if it exceeds the levels of the ambient turbulence might increase their fatigue. Is there an indication on the above? Could that be important to take into account?

Please also note the supplement to this comment:

[revised manuscript text omitted]

---

## Author Comment (AC1) · 25 May 2020

Dear Editor, dear Reviewer,

thank you very much for your comments and for the time dedicated to this work.

In the following we go through your *comments* and provide, for each one, both our *answers* and the *actions* we took to comply with your suggestions.

We will welcome any further comment and suggestion from your side.

Best regards,

The Authors

**Reply to Reviewer #1**

Comment
This is an interesting topic, and the conducted research and associated conclusions are worthy of publications. But the clarity of the writing is quite poor. There were many sentences that i fundamentally could not understand, and most of the paper has rather awkward English. It really needs a review and edit by a native speaker. Once that is complete, i would be willing to re-review, and i think that will go smoothly since the actual research is quite interesting.

Answers
We thank the Reviewer #1 as he or she has acknowledged that content and findings of our research is interesting and worthy of publication. With the aim of replying to the other Reviewers, we modified the manuscript, and, at the same time, as suggested by Reviewer #1, we reviewed the text so as to improve its clarity and grammar.

Actions
We revised the whole text also from the point of view of language, grammar and typos.

**Reply to Reviewer #2**

Comment
This paper is in a very relevant and active research area of wind farm control. Many studies have been published in the last decade focused on the benefits of different wind farm control strategies, such as wake redirection and induction control. However, there has been only limited attention on the impact of wind farm control on the loads of individual wind turbines.
These papers have not been cited well enought in the introcution section, and the actual contribution of the paper is not clearly motivated by building up on what is already known.

Answers
Thank you for this comment. Regarding the bibliography, we reckon we may have missed some important references, especially the most recent contributions. According to your feedback, we have updated the survey of the state-of-art by including additional recent sources. We also tried to further clarify the scope of this paper in terms of its objectives and innovative contents.

Actions
We have improved the state-of-the-art analysis related to this work and hence added the following references to the introductory discussion:
- Boorsma, K., 2012
- Damiani et al., WES 2018
- Zalkind and Pao, ACC 2016
- Mendez Reyes et al., WES 2019
- Kanev et al., Wind Energy 2018

We also edited the Introduction and the Methodology sections to accommodate the new references and to better clarify the scope of this work.

Comment
My most important comment, however, is the scope of the study. Fatigue loads in a wind farm cannot be evaluated on a single wind turbine level. A single turbine operating with yaw misalignment in free stream as first in a row may increase its loads for some yaw misalignments. Behind the turbine, however, the downstream turbines might benefit reduced fatigue loading as the wake is moved away from them. Considering one single turbine in free stream does not provides completely no basis for drawing conclusions on the overall fatigue loading. Instead, all wind directions must be considered in a detailed analysis, including the proper wind speed and direction distributions.

Answer
We completely agree on your consideration that a proper fatigue assessment should go far beyond the limited framework considered here. In particular, we are fully aware that fatigue might be reduced on downwind turbines and that an overall evaluation of the impact of fatigue on the wind farm requires much more detailed analyses.
However, what we did in this paper was to treat fatigue according to the current standards. This means that, for example, in our analyses the actual TI depends on the wind speed rather than being a fixed value. We also assume that the chosen WFC is constantly activated so that the worst-case scenario is considered. In our opinion, this is a sound approach to evaluate the impact of fatigue on the individual wind turbine as part of our parametric analyses. Besides that, our results clearly show that a non-neglectable contribution to the fatigue loads *with* WFC is related to the increase in the rotor mass so that fatigue is actually an *indirect* consequence of the WFC.

Actions
We have tried to better explain our point of view on fatigue in the "Introduction" and "Methodology" sections.

Comment:
Therefore, I cannot approve the paper unless one of the following modifications are made:

• fatigue loads are completely removed and the discussion focusses on ultimate loads only (which I would recommend), or

• it is clearly stated that the fatigue load analysis performed is too limiting to draw conclusions about the overall impact on the lifetime fatigue loading on any turbine in the wind farm, and that further research is needed to study these effects. In doing so, the presented results are to be compared to earlier results by other researches. Notice that the fatigue results, as they stand now, will still not provide much added value in terms of what is already published

Answers
As we discussed in the previous comment, we are aware that our description of fatigue does not allow to draw general conclusions about the whole wind farm. However, once again we believe that this procedure is fully appropriated to conduct our parametric and design analyses. As mentioned, our description of fatigue is compliant with current standard and allows to characterize fatigue loads on the individual wind turbine adequately. In this paper, in fact, we did not focus on other turbines, as this will be the topic of a future continuation of this work. Since our ultimate goal is to quantify the redesign effort, we think fatigue is an important part of this work, as the final structural design is heavily dependent on both ultimate and fatigue loads. On this basis, we have tried to follow your second suggestion, that is, we tried to provide a better framing of our fatigue analyses by discussing its limitations and possible alternative approaches.

Actions
As written in the previous two points, we have included a survey of other research in the field of fatigue analysis of wind farm and provided a qualitative comparison of the main assumptions and limitations. We also clarified why our current approach is, in our opinion, fully compatible with our scope. Moreover, we added a sentence to the scope of the paper to emphasize that the investigation in terms of ultimate loads and maximum tip deflection represents the major source of novelty of our paper.

Comment
The part of the paper related on ultimate loads is relevant and novel. In my opinion, this should be the focus of the paper. Also, here, the story needs to be put in the right perspective. You can't just claim that because of the sensitivity type of analysis, the results and conclusions can be generalized to any wind turbine type. This is obviously not true, as extreme loads depend on many aspects, such as turbine aerodynamics properties, wind conditions, and (supervisory) control system. This needs to be mentioned.

Answers
We fully agree on this. In fact, we specified multiple times that the findings of this study only apply to the turbine under consideration, and that the results hold true for a rotor design constrained by tip deflection. As we reckon that some doubts can still persist, we tried to further clarify this in the Conclusions and we also have updated the title of the paper to further stress that the analyses have been done on a 10MW wind turbine.

Actions

We have rewritten wide parts of the conclusion stressing out that the findings are limited to the case under investigation. We also changed the title to better reflect the scope.

Comment
The part on blade redesign is obscuring the focus of the paper, and given its current length, I propose to completely remove Section 5. Same holds for Sections 3.2 - 3.3 (also related to redesign), although Table 1 should remain detailing the load cases.

Answers
As we discussed in the paper we believe that, given the scope of this study, the part on redesign is fundamental. It gives a quantitative evaluation of the redesign effort required by the adoption of a WF controller. Although these results can be preliminary or limited by all the considerations above, to our knowledge it is the first time a realistic mass increment has been computed with a state-of-the-art design tool. We acknowledge, however, that the section describing the design procedure probably gives too much information and could be shortened.

Actions
After carefully discussing your suggestion among us, we decided eventually not to remove the design. However, we have shortened significantly the description of the design procedure. Specifically, we have merged sections 3.1, 3.2, 3.3 into a single section to condensate those parts.

Comment
Furthermore, there are lots of typos and other errors, please revise the language toroughly before resubmitting.

Answers
Thank you for pointing this out. We will have the manuscript proofreaded before the new submission.

Actions
We revised the whole text also from the point of view of language, grammar and typos.

**Reply to Reviewer #3 (V. A. Riziotis)**

Comment
The paper assesses the effects of two particular wind farm control methods i.e. WR and DIC on the ultimate and fatigue loads of the turbines that the wind farm controller commands to take control action (usually the ones of the front rows of a wind farm). Then they re-design the rotor blades for the DIC one which proves to be the most critical, both in terms of strength and fatigue.
In the reviewer opinion the paper addresses a very interesting and important topic and deserves publication after some revision is made to the original text, including the re-polishing of the language (among others some suggestions for language corrections are given in the supplement pdf).

Answers
We thank the Reviewer because he provided a concise but precise description of the research. We have fixed and improved the language throughout the paper.

Actions
We revised the whole text also from the point of view of language, grammar and typos.

Comment
Please see below my main concerns and points to be further elaborated in the revised text:
1) By reading the title, very large expectations are created to the reader, that the actual impact of the wind farm control on the design loads will be assessed. However, as explained in section 1 and 2, the work turns out to be a parametric study of the effect of i) yaw misalignment and ii) periodic collective pitch angle variation on design loads. Finally the re-design of the blade is only needed and performed for the latter. In order to support originality of the proposed work I would recommend the authors to try to link the conditions scanned in 4.1 and 4.2 with the actual expected conditions in the occasion of wind farm control.

Answers
We acknowledge that the title could be misleading and, after an internal discussion, we will propose to the Editor to modify it to give a better representation of the contents, the final one should be "Evaluation of the impact of wind farm control techniques on fatigue and ultimate loads for a 10MW wind turbine". As you point out, the scope is to provide an evaluation of the impact of (some) WFC on the driving loads. In this view, we believe our methodology could go in the above direction. In fact, the preliminary parametric analysis is fundamental to study the behavior of the chosen WFC and to identify which settings are the most demanding. Clearly this activity could be extended to other WFC techniques, but we decided to focus only on PCM and WR because they are by now common and widespread techniques. As for the redesign, we only present the PCM because PCM has the strongest impact on the sizing loads (see Table 2). Regarding your last suggestion, our goal was to provide a general analysis by considering all the possible conditions and without referring to a specific case (wind farm layout, wind rose etc…). Clearly, we are probably working under an over-conservative assumption but, in our opinion, this is helpful in assessing the general impact of certain WFC before dealing into case-specific scenarios. Additionally, the map between the conditions scanned for different yaw angles and PCM frequencies can be performed following the work of Zalkind and Pao, ACC 2016. This reference, not present in the original version of the paper, is now mentioned in the manuscript to further stress this point.

Actions

To clarify the scope of this study, we better clarified our assumptions and tried to better describe our general approach. We also made it clear that the conclusions have some generality but the actual impact of any WFC should be evaluated on a case-by-case dedicated activity.
To avoid any mismatch between the title and the contents of the paper, we also modified the title.

Comment
2) There are several independent studies which indicate that overall, yaw misalignment, positive and negative increase the DELs ofthe flapwise bending moment (especially as it increases towards -/+20-30deg). In the reviewer opinion, some more convincing explanation of why this is not predicted by the present work must be given (e.g. some time series plot explaining this reduction etc.).
A reference on the yaw correction model used in Cp-lambda is also missing. This is very important in order to accurately predict loads variations in yaw. Moreover, it is stated that the DELs decrease because of the reduction in the mean value of the load. It would be nice to provide the formula of the DEL calculation used by the authors, as the standard one, that the reviewer considers, does not involve the mean load value but only the ranges of load variations. Furthermore, it would be nice to provide the DEL reference frequency and exponents used in the different components DELs calculation (i.e. blades and tower).

Answers
We modified section 4.1 to clarify and/or correct the text as requested. In particular:
1. There are many factors that significantly affect the fatigue loads, and the final result depends on how all these factors mix together. For instance, the yaw misalignment reduces the effective wind blowing on the rotor (i.e. the wind perpendicular to the rotor disc). The turbine controller, in the below-rated region, reacts to this, by trimming the machine at lower rotor speeds, pushing at a lower frequency all the deterministic loads (i.e. those due to wind shear, blade weight, etc.). Therefore, this leads to a reduction of the loading cycles, considered in the fatigue computation. But, at the same time, the yaw misalignment, by itself, generates a crossflow which, in turn, creates an advancing/retreating blade effect. This increases the load oscillations on the blades and hence the DELs. Finally, this advancing/retreating blade effect is made not-symmetric with respect to the yaw angle due to the wind shear. In fact, when the advancing blade is on the top or on the bottom, it experiences a different wind speed value due to the wind shear. For these reasons, it is extremely difficult to envision a priori the effect of yaw misalignment on fatigue loads. It is even more difficult to compare our results with those already published, as seldom the conditions used for computing fatigue are so detailed. But the results we obtained are compatible with what has been already seen in the bibliography. For instance, a reduction of fatigue loads for positive misalignment was already obtained in other works (Ennis, TORQUE 2018; Boorsma, ECN technical report 2012). These aspects are better explained in the revised text, along with the citation of some results present in literature.
2. Some sentences have been added to clarify the computations of loads of our simulator in the case of yaw misalignment.
3. We used the standard formula for DEL calculation. The reviewer is right, as the sentence "in a misaligned configuration the flow velocity perpendicular to the rotor disk is lower, and entails lower loads on the machine" (section 4.1) is imprecise and has been reformulated.
4. DEL frequency and exponents have been reported in the text as suggested.

Actions
This section of the paper has been modified according to the above answers. We have added a comment to figure 3-left and a comment to the yaw model as well as numbers for the DELs calculations.

Comment
3) With respect to DIC the results are as expected. One point that perhaps needs some more attention is to give an indication of how far these perturbations in the wake flow generated by the upstream turbines, travel. Of course they facilitate mixing in the wake but do you also expect a fast decay of the low frequency coherent

fluctuations introduced by collective pitching?  These may cause additional low frequency wind inflow variation to the downstream turbines which if it exceeds the levels of the ambient turbulence might increase their fatigue.  Is there an indication on the above? Could that be important to take into account?

Answers
We totally agree with the reviewer. Clearly, the analysis suggested by the Reviewer is extremely interesting but requires a CFD study, which is out of the scope of the present paper. However, we have already started a dedicated investigation on the impact of DIC on downstream turbine aeroservoelasticity based on SOWFA+Fast simulations. A conference paper is under review, while a journal paper is in preparation. Preliminary, we can say that such an impact may be non-negligible in terms of turbine and trimmer performance, as actually inferred by the reviewer. We added some sentences in section 4.2 to stress this fact.

Actions
We added modified section 4.2 according to our previous answer.

Comment
Please also note the supplement to this comment:

Answers
Thank you. We have addressed the comments provided in the supplement document

Actions
Modified the paper according to the comments provided in the supplement document.

---

## Referee Report (RR1)

[referee-annotated manuscript omitted]

---

## Referee Report (RR2)

**Review of "Evaluation of the impact of wind farm control techniques on fatigue and ultimate loads for a 10 MW wind turbine" by Alessandro Croce, Stefano Cacciola, Luca Sartori, and Paride De Fidelibus.**

This paper falls in two parts. The first part presents considerations of the impact of farm control on an individual wind turbine quantified in term of fatigue and ultimate loads as well dynamic blade response under extreme conditions using a simplistic showcase. Two examples of active wind farm wake control are in focus - wake re-direction and dynamic induction control. The second part uses the results obtained in the first part to perform a wind turbine blade re-design with the purpose of ensure structural integrity under increased loading of the solitary wind turbine investigated in the first part.

The topic is interesting and relevant to the community, but the approach - in its present form - is too simplistic to give meaningful/useful results. Roughly speaking the paper is composed of two 'half papers' (cf. more detailed comments below), and would benefit from selecting a more focused topic (e.g. first part) and then work this trough in more details, however, still using a simple showcase. In its present form, the paper is not ready for publication in the Wind Energy Science journal. Detailed comments/suggestions are given below.

**General comments**

- *Heading*: Due to the simplicity of the selected showcase - where only loading of a single wind turbine in a *non-waked* inflow condition is considered - the wording 'wind farm' should be toned down, and the heading modified to e.g.: Aspects of the impact of active wake control on fatigue and ultimate loads for a 10 MW wind turbine.
- *Scope*: Following the comments in the introduction, the paper could benefit from narrowing down the scope to e.g. consider only the first part (i.e. impact of active wake control on wind turbine loading and dynamics), and then treat this topic more elaborate. This can e.g. be done simplistically by considering the smallest possible wind farm - a two-turbine setup - allowing for analyses of important wind farm load and production characteristics such as wind turbine *spacing* and wind turbine *offset* relative to the mean wind direction (i.e. full wake, partial wake cases).

  The premise for the wake re-direction case in the first part of the paper is that wake effects of downstream wind turbines can be completely mitigated. This is usually not the case. The premise is established based on a stationary flow model (cf. Fig. 1). Stationary modeling of the inflow field makes sense for *production* prediction. However, for *load simulations* wake dynamics is important, and un-steady modeling of wind farm flow fields is needed. This stochastic dynamics comes on top of the static flow illustrated in Fig. 1, and the consequence is that, even in the case of considerable reduced wake production losses, non-neglectable wake loading of downstream turbines may occur. The 'efficiency' of wake re-direction is usually of the order of 0.5D at approximately 10D downstream - somewhat less for densely spaced wind farms. Therefore analyses of different wind farm spacings should be performed.

  The second part of the paper (Section 5), which is which is anyway limited to the PCM control strategy with parameters apparently somewhat arbitrary - or at least not

motivated in terms of cost efficiency - could be the topic for another paper based on more detailed findings from a focused first part of this paper.

- o **Wind speed limitation**: In the present study, the wind farm controller (i.e. the wake steering) is only active for wind speeds lower than 15m/s. This makes perfect sense for production optimization, but for *load mitigation* of e.g. a system consisting of two wind turbines, where one is operating in the wake of the other, this is less obvious and should be motivated. It is a weakness of the paper that only possible increased loading of the wake generating wind turbine is considered without considering the possible load reduction of a downstream wind turbine.
- o **Dynamic induction control**: In dynamic induction control, the magnitude of the thrust force of an upstream turbine is varied, which leads to increased power and thrust variations. This in turn negatively impacts power quality and fatigue loading of the wind turbine subjected to this type of active wake control. Therefore, maybe a comparable approach - the helix approach - could be considered. Investigations (https://onlinelibrary.wiley.com/doi/full/10.1002/we.2513) show that this approach leads to enhanced wake mixing (like dynamic induction control) with minimal power and thrust variations.
- o **Ultimate loading**: The paper claims to deal with wind farm ultimate loading but in reality only deals with ultimate loading of a solitary wind turbine under various (wind farm related) operational conditions. Addressing wind farm ultimate loading requires in addition extreme loading of a wind turbine operating in a waked flow field to be considered.
- o Choose wind farm or WF throughout.
- o Choose wind farm control or WF control or WFC throughout.

**Editorial, semantics and minor comments**

- P.1: "…production, possibly weighted with the wind Weibull": Inclusion of wind direction and wind speed pdf's are a minimum for a trustworthy estimate wind farm production.
- P.2: A number of studies dealing with the impact of WF control on WT fatigue loading (Cardaun et al., 2019; Ennis et al., 2018; White et al., 2018; Boorsma, 2012; Damiani et al., 2018; Zalkind and Pao, 2016) is mentioned, but the results of these studies are not shared with the reader.
- P.2: … the possible increase in machine loading induced by wind farm control: I would actually expect that wind farm control aiming at increased production in most cases will lead also to load mitigation due to less severe wake effects.
- P.3: To the best Authors → To the best of the authors
- P.3: All analyses are performed in this work → In this work, all analyses are performed
- P.3: organized according to the following plan → organized as follows
- P.3: … wind farm controller (WFC) has on the single wind turbine (WT). Why not introduce these acronyms in the introductory section, where 'wind farm control' and 'wind turbine' are also mentioned.
- P.3: WFC is highly site specific. YES, INDEED!!
- P.4: in those conditions which → in conditions which
- P.4: certainly involved in wake interaction → certainly influenced by wake interaction
- P.4: Figure 1 on left → Figure 1, left panel,
- P.5: has to yaw of significant yaw angles → has to yaw significant yaw angles

- P.5: … in order to provide an analysis of general validity: I'm not convinced this is possible for a case study consisting of very few WTs. This statement needs at justification/motivation.
- P.5: if only because a machine → because a machine
- P.5: In this work we assume → In this work we use
- P.7: For example, it may happens that, → For example, it may happen that,
- P.8: … simplify the analysis and make it of general validity, the farm control is considered active only in a range of wind speeds: This intuitively a good idea for power optimization - less intuitive for load mitigation.
- P.8: speed (i.e. up to a → speeds (i.e. up to a
- P.9: inverse SN-curve slope → inverse SN-curve slope (i.e. Wöhler exponent)
- P.9: only for wind speed lower than 15m/s the wind farm controller (i.e. the wake steering) is active: Justify that this is a sensible choice for WFC when also WT loading of downstream WTs is taken into control - see e.g. IOP Conf. Series: Journal of Physics: Conf. Series 1102 (2018) 012019 doi :10.1088/1742-6596/1102/1/012019 in which optimal yaw strategies are defined for above rated wind regimes, where no power loss occurs.
- P.9: regardless to the yaw → regardless of the yaw
- P.9: lacks in generality → lacks in generality and will be highly dependent on the site wind rose
- P.9: … rotated of an angle → … rotated an angle
- P. 12: ((Munters and Meyers, 2017, 2018)) and → (Munters and Meyers, 2017, 2018) and
- P.12: via a wind tunnel experimentation → in a scaled wind tunnel experimental campaign
- P.13: … larger pitch amplitude and Strouhal number → larger pitch amplitudes and Strouhal numbers
- P.14: may arrive to 20% and 30% → may amount to 20% and 30%
- P. 15: PCM: … impact of such a pulsating flow with downstream machines can be significant in terms of turbine loads and aero-servo-elasticity. This particular study, out of the scope of the present paper. BUT IT SHOULDN'T - same comment as for the wake re-direction!
- P.15: nodding ant the → nodding and the
- P.17: no matter of the wind direction an TI → no matter of the wind direction and TI
- P.18: How does the increased PCM induced loading balance with the potential increase in power production in a COE context? This is the crucial question every wind farm owner will address - why should they otherwise accept the increased loading shown in Table 2? It makes no sense to include the detailed analyses presented in Section 5 as long as this basic question isn't addressed. As an illustration, it is stated that "on the other hand, only the PCM with amplitude of 2 deg., the impact of PCM and WR becomes comparable" - yes correct, but what is then the sensibly choice in a WF control context … and why?
- P.18: different wind farm controls have → different wind farm control strategies have
- P.18: skip Sec. 5 (which is anyway limited to the PCM control strategy with parameters apparently highly arbitrary - or at least not motivated in terms of cost efficiency) and

elaborate more on Sec.4 to approach more complete picture of WF loads associated with active wake control.

- P.19: For what concerns → Concerning
- P.19: curve which → curve, which
- P.19: (left)) which → (left)), which
- P.20: Baseline → baseline
- P.20: Then, an important → Thus, an important
- P.21: Baseline → baseline
- P.21: Redesign PCM → re-designed PCM
- P.22: Baseline → baseline
- P.22: Redesign PCM → re-designed PCM
- P.23: Standards and are not site-specific → Standards, and which are not site-specific

---

## Referee Report (RR3)

Review of "Evaluation of the impact of active wake control techniques on fatigue, ultimate loads and rotor design for a 10 MW wind turbine" by Alessandro Croce, Stefano Cacciola, Luca Sartori, and Paride De Fidelibus.

Compared to the previous version of the paper, the paper has improved, but the basic problem remains, which is:

- Since the scope of the paper is to investigate load aspects of WR and PCM wind farm control, the rotor design part of the analysis *assumes* that the front wind turbine is the most critical loaded wind turbine in a wind farm, which is also argued several places in the paper. In my opinion this premise is wrong or at least not documented. There exists lots of evidence both based on numerical simulations and detailed analyses of full-scale data that fatigue loading of wind turbines inside a wind farm is significantly more severe than fatigue loading of solitary (or frontal wind turbines, if you like) wind turbines. Regarding ultimate loading, which is the design driving load scenario in the paper's section 5 analysis, I'm not aware of any systematic investigation documenting that ultimate loading of the frontal wind turbine is more severe that ultimate loading of a wind turbine inside the wind farm. If such documentation exists that can support your premise of considering 'the worst possible scenario', it must definitely be referenced, as it is needed to justify section 5 the rotor design part.
- Following the rationale above, it is in turn not justified/documented that ultimate loading in general is more critical for a wind farm turbine than fatigue loading, which is a comprehensive study of its own. What has been demonstrated is, that for the two selected active wake control approaches, a frontal wind turbine is more severely loaded by extreme external events that by fatigue, when evaluated in terms of the IEC standard load cases. However, as all wind turbines are considered of the same type in the wind farm (which is indeed a reasonable assumption), a detailed rotor design study is too premature, unless documented that ultimate loading of a frontal wind turbine is design driving for all turbines in a wind farm.
- Another aspect as I read the design study is that the specified PCM control is included in all relevant DLCs for the entire lifetime of the turbine. This is somewhat conservative, as a particular frontal turbine will only be a frontal turbine, and thus operate under PCM control, for part of its lifetime.

In my opinion the consistency of the paper will be considerably improved by only including the first 4 sections and a conclusion. The first 4 sections constitute a load study - fatigue as well as ultimate - of a frontal turbine operating under WR and PCM wind farm control. A possible paper title could then e.g. be: "Evaluation of the impact of selected active wake control techniques on fatigue, ultimate loads and rotor design for front row 10 MW wind turbine".

**Editorials:**

P.1: All works suggested -> All works suggested

P.2: aggregate damage equivalent loads was -> aggregate damage equivalent loads were

P.4: depend on many factors, such as the farm layout, the wind distribution and rose, the turbulence intensities. -> depend on many factors, such as the farm layout, the wind distribution and rose as well as the turbulence intensities. P.4: and/a fact the a > and/ar fact the

P.4: and/o feel the -> and/or feel the

P.6: through 1D geometrically exact -> through a 1D geometrically exact

P.7: a convergence solution is found -> a converged solution is found

P.15: The change of thrust, in facts, results -> The change of thrust, in fact, results

---

## Author Response (AR2)

Dear Editor, dear Reviewers,

thank you very much once more for your comments and for the time dedicated to this work.

In the following we go through your *comments* and provide, for each one, both our *answers* and any *actions* we took to comply with your suggestions.

We welcome any further comment and suggestion from your side.

Best regards,

The Authors

**Reply to Reviewer #2**

**Suggestions for revision or reasons for rejection**
None

Answers
We thank again the Reviewer #2 for the time dedicated to this work.

Actions
None.

**Reply to Reviewer #3 - Vasilis A. Riziotis**

**Suggestions for revision or reasons for rejection**
The paper has significantly improved since the first submission.
The authors convincingly responded to all my comments/questions.
Moreover, the language has been greatly improved. However, I still find small grammar/syntax errors here and there.
A final careful reading is recommended. I indicated some of the errors in the attached pdf.

Answers
Thank you for this comment. We went through the document and correct the typos/errors highlighted by the reviewer.

Actions
We revised the whole text also from the point of view of language, grammar and typos.

**Reply to Reviewer #4**

**Suggestions for revision or reasons for rejection**
The topic is interesting and relevant to the community, but the approach - in its present form - is too simplistic to give meaningful/useful results. It's difficult to justify that a paper dealing with wind farm control is build around a 'showcase' consisting of only one wind turbine. Thus, the paper will benefit from taking a more focused approach - e.g. to be condensed to the first part of the paper- and then work this through in more details, however, still using a simple showcase (consisting of minimum two wind turbines)..

Answers
We thank you again the Reviewer for her/his comment. We fully understand the Reviewer's vision, but the goal of this paper is to look at the wind farm control problem from a wind turbine design point of view. As it will be clear from the reply to the next comments, this implies, according to the International Regulations, that one has to consider the worst possible scenario. From a practical point of view, this leads to a formulation of the problem which does not eventually require (at least for the purpose of our investigation) a simulation of the whole farm flow. At the same time, in line with the reviewer's comment, some of the authors assessed the problem from the point of view of wind farms in another work presented at TORQUE2020 and soon to be published in a journal paper. What is described in this paper remains, in our opinion, of fundamental importance to show the impact of the wind farm control techniques on the design of the wind turbine.

Actions
No action at this point.

Reviewer comment:
This paper falls in two parts. The first part presents considerations of the impact of farm control on an individual wind turbine quantified in term of fatigue and ultimate loads as well dynamic blade response under extreme conditions using a simplistic showcase. Two examples of active wind farm wake control are in focus - wake re-direction and dynamic induction control. The second part uses the results obtained in the first part to perform a wind turbine blade re-design with the purpose of ensure structural integrity under increased loading of the solitary wind turbine investigated in the first part.
The topic is interesting and relevant to the community, but the approach - in its present form - is too simplistic to give meaningful/useful results. Roughly speaking the paper is composed of two 'half papers' (cf. more detailed comments below), and would benefit from selecting a more focused topic (e.g. first part) and then work this trough in more details, however, still using a simple showcase. In its present form, the paper is not ready for publication in the Wind Energy Science journal. Detailed comments/suggestions are given below.

Answer:
The Reviewer accurately described the main content of the paper, but we share only partially her/his opinion that the manuscript is composed by two "half papers". On the one hand the paper is certainly composed of two parts, which, taken individually, could be extended, but on the other hand we believe that we can have a correct overview on the impact of wind farm control in terms of wind turbine design only by putting these two parts together as we have done. In fact, from a design point of view, one is interested in evaluating the machine loading in different scenarios according to Standards and then in designing the machine considering the worst possible conditions. The mainly focus of this research is exactly this one, i.e. to evaluate the possible

impact of wind farm control strategy(ies) on the rotor design, an important industrial and research topic which, according to our best knowledge, has been never addressed within the current literature. In this design framework, it is possible to understand our choice of considering, in this first paper on this topic, the single upwind wind turbine for the investigation, without detailing possible (to be carefully evaluated) reduction in downstream machine loading. This point, which, as far as we understand, represents the major concern of the Reviewer, will be detailed in the rest of this point-by-point reply.

Action:
No action at this stage.

Reviewer comment:
Heading: Due to the simplicity of the selected showcase - where only loading of a single wind turbine in a non-waked inflow condition is considered - the wording 'wind farm' should be toned down, and the heading modified to e.g.: Aspects of the impact of active wake control on fatigue and ultimate loads for a 10 MW wind turbine.

Answer
This comment is similar to one of Reviewer #3 during the first round of review. As in that occasion, we are open to modifying the title not to create a misleading expectation in the reader

Action:
We propose to the editor to change the title in "Evaluation of the impact of active wake control techniques on fatigue, ultimate loads and rotor design for a 10 MW wind turbine"

Reviewer comment (Scope - 1):
Following the comments in the introduction, the paper could benefit from narrowing down the scope to e.g. consider only the first part (i.e. impact of active wake control on wind turbine loading and dynamics), and then treat this topic more elaborate. This can e.g. be done simplistically by considering the smallest possible wind farm - a two-turbine setup - allowing for analyses of important wind farm load and production characteristics such as wind turbine spacing and wind turbine offset relative to the mean wind direction (i.e. full wake, partial wake cases).

Answer:
Here again, we strongly believe that the paper should have these two lungs, impact on turbine loads and on rotor design, and a single hearth, evaluating wind farm control effects at turbine level from the perspective of the design compliant with current Standards. We believe the new proposed title could emphasize exactly this.
Dealing with the fact that the analysis should consider also downstream (waked) turbines, this comment was similar to one of Reviewer #2 in the first round of review. Of course, we agree with this consideration, but the focus of the paper is slightly different. We certainly know that wake redirection may have a positive impact on downstream machine loading, but from the design standpoint, one is interested in the worst possible case, which happens, as showed in this work, when the turbine is upstream. Similarly, an analysis about turbine spacing, although extremely interesting, is not necessary in this paper. In fact, any turbine is designed once and then is supposed to operate in different geographical locations, different farms, and different positions within the same farm.

Action
We stress even more this concept by adding the following sentence in the introduction immediately after the declaration of the scope of the paper.

"As the design of a new rotor has to be carried out according to the Standards and has to consider the worst possible scenario, the analyses in this paper will focus on the isolated upstream machine, under different farm-related operating conditions. In fact, as it will be pointed out in detail in Sec. 2, in the simple case of wake redirection control, upstream turbines are more prone to the negative impacts of the farm control, e.g. those entailed by operations at large yaw angles, while the downstream ones will possibly experience all the advantages, e.g. lower turbulence and lower wake impingement with respect to the case without wind farm controllers.

Clearly, in a single farm, there is a subset of machines which most often see a clean flow, i.e. the outer ones exposed according to the most probable wind direction, and another subset of turbines, the inner ones, which sometimes see a waked flow. In this scenario, it is certainly interesting to evaluate a possible usage of partially customized or totally different turbines in a single farm, depending on the specific machine location. In such a case, the turbines proposed for the innermost farm locations may be characterized by more competitive designs thanks to the farm control. Although extremely interesting, this idea falls out of the scope of the present paper."

Reviewer comment (Scope - 2):
The premise for the wake re-direction case in the first part of the paper is that wake effects of downstream wind turbines can be completely mitigated. This is usually not the case. The premise is established based on a stationary flow model (cf. Fig. 1). Stationary modeling of the inflow field makes sense for production prediction. However, for load simulations wake dynamics is important, and un-steady modeling of wind farm flow fields is needed. This stochastic dynamics comes on top of the static flow illustrated in Fig. 1, and the consequence is that, even in the case of considerable reduced wake production losses, non-neglectable wake loading of downstream turbines may occur. The 'efficiency' of wake re-direction is usually of the order of 0.5D at approximately 10D downstream - somewhat less for densely spaced wind farms. Therefore analyses of different wind farm spacings should be performed.

Answer:
Figure 1 is not to be viewed as a premise for wake redirection case, but rather as a simple investigation to derive some reasonable pieces of information related to the region of activation of farm control in terms of speed and TI, and compare such conditions with those prescribed by Standards.
We agree with the Reviewer that loading of impinged rotor is to be evaluated considering a dynamic wake. In fact, we have already started a dedicated analysis and some preliminary results are already available (see https://doi.org/10.1088/1742-6596/1618/6/062033).

Action:
To stress this, we added the following sentence in the conclusions.
"In terms of extensions of the proposed work, the evaluation of the impact of farm control on ultimate and fatigue loads of downstream turbines is certainly an interesting topic, which deserves dedicated analyses with more sophisticated tools than those used in the present work to simulate the wind farm flow, e.g. CFD or dynamic wake models."

Reviewer comment (Scope - 3)

The second part of the paper (Section 5), which is which is anyway limited to the PCM control strategy with parameters apparently somewhat arbitrary - or at least not motivated in terms of cost efficiency - could be the topic for another paper based on more detailed findings from a focused first part of this paper.

Answer
We believe that the paper, without the design part, would be incomplete according the very scope of the work, as we detailed in the answer to a previous comment.
PCM control parameters have been selected according to the preliminary sensitivity study. Moreover, the range of variation of frequency and amplitude, has been selected from a set of experimental activity in wind tunnel presented in a previous paper (see https://doi.org/10.5194/wes-5-245-2020).

Action:
A sentence at the beginning of Sec. 4.2.2 is modified and now reads:
"Different combinations of amplitude $A\_PCM$ and Strouhal number $St$ were considered: the range in amplitude was set between 1 and 4 deg, whereas the range of Strouhal between 0.2 and 0.5., according to the findings of an experimental campaign in wind tunnel (see Ref. Frederick et al., 2020)"

Reviewer comment:
Wind speed limitation: In the present study, the wind farm controller (i.e. the wake steering) is only active for wind speeds lower than 15m/s. This makes perfect sense for production optimization, but for load mitigation of e.g. a system consisting of two wind turbines, where one is operating in the wake of the other, this is less obvious and should be motivated. It is a weakness of the paper that only possible increased loading of the wake generating wind turbine is considered without considering the possible load reduction of a downstream wind turbine.

Answer
As written in one of the previous replies, our choice is to look at the wind farm control problem from the design standpoint, and, accordingly, we consider the worst possible scenario. We do not believe that this is a weak approach, and this is particularly evident for ultimate loads. Consider the case of wake redirection and, as usual, all turbines of the same type in the farm. We may have an increase of a specific ultimate load in the upstream turbine, and a reduction in downstream one(s). Since all turbines are of the same kind and we have to select the highest load, the design process results to be blind respect the positive impact in the downstream machine.

Action
No action.

Reviewer comment:
Dynamic induction control: In dynamic induction control, the magnitude of the thrust force of an upstream turbine is varied, which leads to increased power and thrust variations. This in turn negatively impacts power quality and fatigue loading of the wind turbine subjected to this type of active wake control. Therefore, maybe a comparable approach - the helix approach - could be considered. Investigations (https://onlinelibrary.wiley.com/doi/full/10.1002/we.2513) show that this approach leads to enhanced wake mixing (like dynamic induction control) with minimal power and thrust variations.

Answer
We knew and had already read the suggested paper. Interestingly, the helix approach paper and ours were submitted to two different journals exactly the same day. Hence, we could not consider the helix approach in the development of this work. All in all, we do not expect that helix approach could significantly ameliorate the blade loads. In fact, with IPC the single blade loads may undergo significant changes, while the thrust (being the sum of the thrust of the three blades) may be subjected to a limited modification. On the other

side, the impact of the helix control on the tower, directly connected to the thrust, can be actually smaller than that of collective dynamic induction control. In any case, we would like to thank the Reviewer once again for this further suggestion, which we will certainly look into in more detail in future work.

Action
We inserted a reference to the helix approach paper in the 'Conclusion and outlook' section.

Reviewer comment:
Ultimate loading: The paper claims to deal with wind farm ultimate loading but in reality only deals with ultimate loading of a solitary wind turbine under various (wind farm related) operational conditions. Addressing wind farm ultimate loading requires in addition extreme loading of a wind turbine operating in a waked flow field to be considered.

Answer
See our answer to a previous comment.

Action:
No action required.

Reviewer comment:
Choose wind farm or WF throughout. Choose wind farm control or WF control or WFC throughout.

Answer
We agree, and since in some statements we think it is more useful to use the full expression rather than the acronym, we have removed the latter from the text.

Action
We removed the acronyms WF and WFC.

Minor Reviewer comments
We skip in the reply of these minor comments, the request for correction of typos. These corrections have been applied directly to the final uploaded version. We would like to thank the Reviewer once again for these suggestions.

Editorial, semantics and minor comments
- P.1: "…production, possibly weighted with the wind Weibull": Inclusion of wind direction and wind speed pdf's are a minimum for a trustworthy estimate wind farm production.
We changed "possibly" with "properly"

- P.2: A number of studies dealing with the impact of WF control on WT fatigue loading (Cardaun et al., 2019; Ennis et al., 2018; White et al., 2018; Boorsma, 2012; Damiani et al., 2018; Zalkind and Pao, 2016) is mentioned, but the results of these studies are not shared with the reader.
We included a brief description of those findings where needed.

- P.2: … the possible increase in machine loading induced by wind farm control: I would actually expect that wind farm control aiming at increased production in most cases will lead also to load mitigation due to less severe wake effects.

See reply to a similar previous comment.

- P.3: … wind farm controller (WFC) has on the single wind turbine (WT). Why not introduce these acronyms in the introductory section, where 'wind farm control' and 'wind turbine' are also mentioned.

See answer to a previous reply.

- P.3: WFC is highly site specific. YES, INDEED!!

Indeed, but according to the vision from a design standpoint we are looking at the worst case, information of general validity and not site-/farm-specific.

- P.5: … in order to provide an analysis of general validity: I'm not convinced this is possible for a case study consisting of very few WTs. This statement needs at justification/motivation.

Again, this sentence can be fully understood, as detailed in the reply to general comments, looking at the problem from the design standpoint. Aspect, which is already written multiple times in the manuscript, including the part which immediately follows the sentence "in order to provide an analysis of general validity", at the end of section 2.

- P.5: if only because a machine → because a machine

Changed as suggested even if the previous sentence was correct.

- P.8: … simplify the analysis and make it of general validity, the farm control is considered active only in a range of wind speeds: This intuitively a good idea for power optimization - less intuitive for load mitigation.

See a reply to a similar previous comment

- P.9: only for wind speed lower than 15m/s the wind farm controller (i.e. the wake steering) is active: Justify that this is a sensible choice for WFC when also WT loading of downstream WTs is taken into control - see e.g. IOP Conf. IOP Conf. Series: Journal of Physics: Series: Journal of Physics: Conf. Series Conf. Series 1102 (2018) 012019 1102 (2018) 012019 doi :10.1088/1742doi :10.1088/1742--6596/1102/1/0120196596/1102/1/012019 in which optimal yaw strategies are defined for above rated wind regimes, where no power loss occurs.

A sentence has been added to clarify this point, together with the reference of the proposed article.

- P.9: lacks in generality → lacks in generality and will be highly dependent on the site wind rose

We corrected the text in "lacks of generality". The problem is not related to the wind rose, but to the fact that the tower has a cylindrical shape and hence evaluating the modification in a direction does not tell the truth about the actual loading of the tower, as explained in the manuscript.

- P.9: … rotated of an angle → … rotated an angle

We changed it in "rotated by an angle"

- P. 15: PCM: … impact of such a pulsating flow with downstream machines can be significant in terms of turbine loads and aero-servo-elasticity. This particular study, out of the scope of the present paper. BUT IT SHOULDN'T - same comment as for the wake re-direction!

See a reply to a previous comment.

- P.18: How does the increased PCM induced loading balance with the potential increase in power production in a COE context? This is the crucial question every wind farm owner will address - why should they otherwise accept the increased loading shown in Table 2? It makes no sense to include the detailed analyses presented in Section 5 as long as this basic question isn't addressed. As an illustration, it is stated that "on the other

hand, only the PCM with amplitude of 2 deg., the impact of PCM and WR becomes comparable" - yes correct, but what is then the sensibly choice in a WF control context … and why?

The question has been posed in the introduction. In the paper, we showed that non negligible increase in blade mass is to be considered if one includes wind farm control in the design process. This is not a go/no go assessment. It is simply an indication to be considered in the farm control optimization and hence in the final cost of energy of the complete wind farm. As explained in the introduction, the problem of wind farm control is the one of minimizing the cost of the energy, which is a more proper merit figure than the maximization of AEP or maximization of AEP with some constraints about fatigue. Clearly, in this paper, we have not solved the entire problem, but have rather provided an indication.

- P.18: skip Sec. 5 (which is anyway limited to the PCM control strategy with parameters apparently highly arbitrary - or at least not motivated in terms of cost efficiency) and
elaborate more on Sec.4 to approach more complete picture of WF loads associated with active wake control.
See reply to a previous comments

- P.20: Baseline → baseline
- P.21: Baseline → baseline
- P.21: Redesign PCM → re-designed PCM
- P.22: Baseline → baseline
- P.22: Redesign PCM → re-designed PCM

"Baseline" and "Redesign PCN" are used in this context as proper names identifying the specific rotor. We believe these names should be maintained as they are.

---

## Author Response (AR3)

Dear Editor, dear Reviewer,

thank you once again for your comments and for taking the time to review our work.

In the following we go through your *comments* and provide, for each one, both our *responses* and the *actions* we have taken to accommodate your feedback in the revised manuscript.

Best regards,

The Authors

**Reply to Reviewer #1**

**Review Comment:**
-Since the scope […]
-Following the rationale above, […]

**Authors Answer:**
We thank again the Reviewer for her/his comment. We fully understand the Reviewer's vision and we agree on these two points. However, one of the main findings of this paper is related to the importance of ultimate loads, along with the fatigue ones, when considering Wind Farm Controllers (WFCs). This important research and industrial problem, to our best knowledge, has never been addressed within the current literature. The parametric analyses, and the following design, are intended for this purpose.
At the same time, as written before, we fully understand the Reviewer point of view, and we tried to further improve this paper as follows:

1. We removed from the title the word "design" and now it reads "Evaluation of the impact of active wake control techniques on fatigue and ultimate loads for a 10 MW wind turbine "
2. We stressed in several sections of the paper that this research topic is aimed at highlighting the direct impact of wind farm controllers on ultimate loads, starting from the "active" wind turbine, i.e., the one where the WFC is operating. As the Reviewer pointed out, this could not be the worst case, but it provides an indication of a lower limit: when one operates the "front" wind turbine with the WFC, the design loads (and the maximum tip deflections) for that wind turbine increases. Again, assuming all turbines of the same type in the farm, the impact of the analyzed control on ultimate loads is at least the one obtained for this first-row turbine. This is an aspect of paramount importance especially for those rotors (or parts of the rotors) for which the design drivers for the main subcomponents of the blade (e.g., the spar caps) are the ultimate loads or the maximum blade deflections, as the case of the reference 10MW turbine analyzed in the paper.
3. We reduced, but not totally removed as suggested by the Reviewer, the section on the design. We also stressed in this section, once more, the focus and the limitations of the proposed approach, as also highlighted by the Reviewer. But, as written before, this section has been (further) reduced and turned into a sub-section in order not to give the impression that this is the main part of the work, but provides additional information on the problem of evaluating WFCs in terms of the design loads.
4. We update some sentences in other sections to be consistent with the above changes.

**Authors Actions**
We update the paper accordingly.

**Review Comment:**
- Another aspect as I read the design study - is that the specified PCM control is included in all relevant DLCs […]

**Authors Answer:**
We do not agree on this point. According to the regulations, the designer has to look for the worst combination of events. Is it true that the WFC operates on the WT for part of its lifetime, but we must consider that the extreme event may occur when the WFC is on. This is the reason why the ultimate loads and the maximum blade deformation may increase.

**Authors Actions**
No action on this point.

**Review Comment:**
- Editorials […]
Authors Answer:
Thank you again. We corrected all these typos.

**Authors Actions**
Typos/errors corrected in the final paper.

[revised manuscript text omitted]

---

## Author Response (AR4)

*Dear Editor,*

*thank you again for your comments and for taking the time to review our work.*

*In the following we go through your comments and provide, for each one, both our responses and the actions we have taken to accommodate your feedback in the revised manuscript.*

*Best regards,*

*The Authors*

**Editor Comment(s):**
-- Concentrate the discussion on ultimate conditions, which is the novelty of your study. This will improve readability and should drastically simplify the discussion, which at the moment is confused and difficult to follow.
- Move the discussion on fatigue to a less prominent position (for example, in an appendix). Better highlight the limits of your results on fatigue (which might increase or decrease, depending on the site and specific location of a turbine in a farm).
- Use design as a way to preliminarily quantify the effects of changes in ultimate loads and tip deflection. Even if fatigue is approximate (see above), still the re-design exercise can have the role of indicating the impact of the changed ultimate states. Clearly indicate the qualitative nature of the re-design and its limits.
- Reflect the new focus also in the title, abstract and introduction.

**Authors Answer(s):**
Thank you again Editor for your comments.
The above points have been addressed in the submitted reports. In particular, the fatigue analysis has been further reduced in the paper (but not completely removed) and, most important, we better highlighted the limitations of this simplified fatigue analyses, also by adding some comments with respect to the present literature. This is also done in the final "design" section, where we tried to better explain the procedure (i.e. the need to have the fatigue loads in the baseline and the need to highlight the impact of the ultimate loads in the re-designed one). Title, abstract and introduction have been consequently updated. The final paper has in red the new sections/sentences.

**Editor Comment(s):**
- Avoid claiming that the study is general because you look at sensitivities. Clearly your results still depend on the specific turbine that you have selected.
- Eliminate statements to the effects that loads are overlooked (abstract), because clearly loads cannot be overlooked by industry given the obvious implications on safety.
- Please review all the discussion from line 128 to the end of Section 2. I cannot follow the logic, and I suggest to eliminate it altogether.
- The discussion on directional dependence of fatigue (from line 259 onwards) is rather obvious, please reduce or consider eliminating.
- "Downwind turbine" might be misunderstood, use downstream or reword.

**Authors Answer(s):**
- We put even more stress on the fact that the results are highly dependent on the model used.
- We added in the abstract the words "in the literature" to better explain that what has been overlooked are not the loads in the Industry, but our proposed analyses on ultimate loads in the scientific literature.

- Section 2 has been simplified and reduced as suggested. Thanks again for this comment, we also now think that this section is more readable.
- Also the discussion on the direction of fatigue loads has been removed. We also think is was rather obvious, even if a lot of papers still do not present the worst fatigue direction, but only the classical fore-aft and side-side.
- "Downwind" has been replaced with "Downstream"

---

## Author Response (AR5)

Dear Editor,

We thank you for acknowledging the improvements that we have done in the last manuscript update, and for suggesting other issues that should be considered to further ameliorate the paper itself.

In the following, you can find a point-by-point reply to all your comments and for each of them the "Actions" that we have implemented to amend the manuscript.

*[Editor] Page 4 "Unfortunately, regulations, in their current status, do not consider yet the fact that a turbine may operate out of the design conditions according to a farm control.". This is not entirely accurate, and you might want to consider this position paper: https://www.windfarmcontrol.info/-/media/Sites/WindfarmControl/Publications/FarmConners-WP2-D2-1-Position-paper-on-WFC-certification_revised.ashx?la=da&hash=E5801009B447D914DD7A566C9FCA9426B4483C9D. In addition to considering wind farm control, the paper also explains well how in-farm inflow conditions are currently addressed by the standards. Material from this paper can help improve the introduction and methodology sections.*

[Reply] We are aware of that Position Paper. Clearly, we could not use it in the development of our work as the FarmConners deliverable was made public in July 2020, while the first submission of our manuscript was done in December 2019. By the way, we agree that it is now opportune to refer also to that position paper (along with a similar Deliverable D4.7 of CL-Windcon project), and update Introduction, Methodology and Conclusion sections.

[Actions] We have strongly modified the Methodology sections and updated the Introduction so as to:

- Clarify that Standards can be used for certifying wind farm controls, even if current Standards "in practice do not cover the wind farm control case explicitly" (cf. Section 2.2 of the Position Paper of FarmConners).
- Stress the fact that Standards suggest different tools to model the in-wake flow, with the possibility to handle load assessment for downstream turbines, especially if fatigue is considered. This, on the other side, offers us the opportunity to emphasize that much is still to be done to quantify the impact on ultimate loads (focus of our paper).
- Clarify, even more, that the focus of the paper is on ultimate loads of front-row turbines. This, although limited for certain aspects, represents a further step towards a comprehensive knowledge of pros and cons of wind farm control technique.

We have also updated the sentences related to the Standards in the Conclusions.

*[Editor] Page 5, "1D geometrically exact …": I understand what you mean, but the sentence might be misunderstood. The model is 3D, and I believe that the term "beam" already implies what you are referring to.*

[Reply] We agree with the editor.

[Actions] The text has been rephrased as "This tool allows one to model the flexibility of blades, tower and shafts through a geometrically exact beam model (Bauchau, 2011), whose sectional structural properties are rendered with fully populated 6 x 6 stiffness matrices."

*[Editor] Page 7: is there any possible failure mode of the wind farm control laws that should be considered? If not, please explain why.*

[Reply] The comment is extremely pertinent, as any possible technology (hard- or soft-ware) may be subject to failures, and wind farm control is not an exception. However, a proper analysis falls out of the scope of the paper. The problem of failures in farm control laws was briefly analyzed in Deliverable 4.7 of CL-Windcon project (cf. http://www.clwindcon.eu/wp-content/uploads/2020/03/CL-Windcon-D4.7-Review-on-standards-and-guidelines.pdf). After reviewing the different techniques and the associated risks, the Authors of that deliverable stated that in case of conflict between individual wind turbine regulator and farm controller, the former should have the priority. This, at least for the goal of our paper, offers a suitable reason not to include farm control failure modes in the analyzed DLCs. Moreover, to some extent, it suggests that farm controllers can be implemented so as to limit the risks connected to their failure. As a final comment, one may also envision that farm controllers may be linked to failure detection systems which would disengage the control itself when needed, restoring the nominal operations based on greedy control.

Clearly, we acknowledge that the issue raised by the Editor deserves a mention within our manuscript.

[Actions] We have included a comment about the wind farm failure modes in Section 3 after Table 1, briefly explaining the problem and the circumstances under which it is reasonable not to include the farm control failure in the analysis.

Moreover, a new sentence related to this point have been included in section Conclusion.

*[Editor] Page 9 (but also elsewhere, wherever you refer to AEP): AEP is not a meaningful KPI in this context, as a turbine will never work with the same misalignment angle for a full year. Power losses for a steering turbine are typically quantified through the cosine law exponent, not AEP (unless you are looking at a specific site, wind rose, farm layout, etc.). The power loss exponent however has a large scatter, and very different values have been reported in the literature. It is doubtful that your (and most other) BEM models would be able to accurately predict this effect, unless high yaw misalignment corrections are implemented and calibrated. I suggest that you simply refer to the literature on this point, which is also not central to the present discussion.*

[Reply] We had inserted the indication of AEP in the previous versions of the manuscript to give an idea on the impact of the control techniques on the production of the single front-row turbine, showing that, for example active wake mixing has lower impact than wake redirection. Hence, for active wake mixing, even a small increase in the downstream turbine production may compensate the loss in the upstream machine and lead to an increase of the overall farm power output.

However, we acknowledge that such indication may be prone to misinterpretations. Hence, we accommodate Editor's suggestion.

[Actions] We have removed the information related to AEP on page 9, Table 2, Table 3. Figure 10 (power curve and power coefficient) have been eliminated and related text have been accordingly modified.

*[Editor] Page 9 (but also elsewhere, wherever you refer to ADC): the previous comment applies also to ADC: even if you use Weibull weighting, it is not reasonable to assume operation for a full year at a given misalignment angle. Additionally, you refer to pitch ADC, but one should first have a discussion on yaw ADC (which is the main concern in practice, since it grows significantly, while pitch ADC will tend to decrease).*

[Reply] Clearly, the ACD for yaw actuator is essential for wake redirection, but it cannot be evaluated by simply looking at the 600-second simulations, considered in our DLC list, where the turbines are not expected to yaw following wind direction changes. Hence, we agree with the Editor that the treatment about ADC may lead to a misleading piece of information. On the other hand, it would be a pity removing the evaluation of pitch-ADC for active wake mixing, as this is the first time in literature that this analysis has been done. Moreover, a more "realistic" indication of ADC increase may be extracted from "full year" ones multiplying the indication that we provided by the percentage time spent with non-null farm control input.

[Actions] We have left the information on pitch ADC in the context of the active wake mixing, clarifying the limitation connected to the fact that we are looking at full year operation with the farm control (page. 14). We have removed the indication of ADC in the rest of the paper. In particular, we have eliminated the left plot of figure 11 (Weibull-averaged ADC) and modified the related comments in the text.

*[Editor] Page 11, end of sect. 4.1: this is also the reason for one-sided wake steering (in addition to the much reduced effects on yaw ADC), which has been consistently used in most of the field tests conducted so far.*

[Reply] We agree with the Editor. We had also mentioned such opportunity in the Methodology section in a slightly broader sense, stating that the authority of the farm control can be limited in all those conditions which can be critical in terms of loading. Clearly, avoiding operations in critical conditions may simply lead to one-sided yaw misalignment or to more complex combinations of misalignment angles, speed, and TI.

[Actions] We have stressed this concept and have referred to previous experiments involving wake steering (e.g., Fleming et Al. Wind Energy Science 4, 273-285, 2019.) both in the methodology section and at the end of section 4.1.

*[Editor] Page 14: please note the strange apparent inconsistency: here you look at the ADC of the driving input (pitch), while for wake steering you look at the ADC of pitch, although the driving input in this case is yaw. As noted above, pitch ADC will decrease, but yaw ADC will very much increase.*

*In addition, here again the WF controller will not be operating all the time, so this KPI does not seem to be very useful. As the work focuses on ultimate loads, I suggest that you eliminate references to AEP and ADC: as for DELs, any analysis of these KPIs requires creating first a reasonable scenario, because WFC is not used all the times. In my opinion, as for DELs in the previous version of the manuscript, references to these aspects unnecessarily complicate the discussion and distract from the main point.*

[Reply] See the reply to previous comments related to AEP and ADC.

[Actions] See the reply to previous comments related to AEP and ADC.

*[Editor] Page 16: you decided to focus only on 2 deg amplitude. But is it enough to achieve a useful increase in wake mixing? Please justify this choice. What happens if we really need 4 deg to achieve some power boost?*

[Reply] This choice is justified by previous experiments in wind tunnel, published in Frederik et al., Wind Energy Science, 5, 245–257, 2020. In that experimentation, the highest power increase was achieved with an amplitude of about 1.7 deg for two different atmospheric boundary layers (cf. Tab 2, Fig 10, and Fig 13 of WES 2020 paper).

Clearly, the active mixing with 4 deg would have a more significant impact on loads, and, eventually, on design. The choice on whether to use it or not, if one really needed 4 deg, would depend on the balance between power boost potentially achieved and detrimental impact on loads. Of course, there is a similar issue for wake redirection-based controllers (see for example, the possibility of employing one-sided wake steering, that the Editor mentioned in a previous comment). All in all, the choice of 2 deg appeared, at least to the Authors, more convenient and supported by previous experience in wind tunnel, as stated before. We do not believe that repeating the design also for 4 deg will give value to the work.

[Actions] We have added a proper comment and the reference to Frederik et al., WES 2020, after Table 2, to better explain the choice of focusing on 2 deg amplitude.

[Editor] Page 17: "… have an impact on ultimate loads, especially on the maximum tip deflection, …". I understand what you mean, but tip deflection is not a load. Please rephrase.

[Reply] We agree.

[Actions] We have rephrased the text.

[Editor] Page 17, lines 389-392: I am not sure I understand the motivation not to consider wake steering in the redesign exercise, since previously you said that it causes a significant increase in tip deflection. When you say "The design process of the baseline generated an optimal solution compliant to all optimization constraints, with a structure mildly different …" are you talking of the redesign accounting for wake steering? Please rephrase, this is unclear.

[Reply] That sentence refers to a preliminary step in the design process consisting in the update of the INNWIND.EU blade, with the aim of providing a compliant baseline, i.e. with a blade designed using the very same constraints and DLCs that we will use in the redesign process that considers the wind farm control. The redesign of the blade including wake redirection is not part of this paper, but is included in a conference one, i.e., Sartori et al. TORQUE 2020. Such article was already included in the list of reference. We may however recall one result (i.e. the increase of blade mass entailed by WR), which may be of interest for readers.

[Actions] We have moved the misleading sentence of page 17, line 389-392, some lines above where the baseline updating is explained. We will also include a sentence about the mass increase after the blade redesign under wake redirection control, taken from ref. Sartori et al., TORQUE 2020.

[Editor] Many sentences are long and a bit convoluted. Please do not rely only on the language editor to improve the language during production, and try to shorten and simplify the text wherever possible.

[Reply] -

[Actions] We have shortened, simplified, corrected, and improved the text of the manuscript.